# Chiral monoterpenes reveal forest emission mechanisms and drought responses

Joseph Byron[1], Juergen Kreuzwieser[2], Gemma Purser[3,4], Joost van Haren[5,6], S. Nemiah Ladd[2,7], Laura K. Meredith[5,8], Christiane Werner[2] & Jonathan Williams[1,9 ✉]

Monoterpenes ($C_{10}H_{16}$) are emitted in large quantities by vegetation to the atmosphere (>100 TgC year$^{-1}$), where they readily react with hydroxyl radicals and ozone to form new particles and, hence, clouds, affecting the Earth's radiative budget and, thereby, climate change[1–3]. Although most monoterpenes exist in two chiral mirror-image forms termed enantiomers, these (+) and (−) forms are rarely distinguished in measurement or modelling studies[4–6]. Therefore, the individual formation pathways of monoterpene enantiomers in plants and their ecological functions are poorly understood. Here we present enantiomerically separated atmospheric monoterpene and isoprene data from an enclosed tropical rainforest ecosystem in the absence of ultraviolet light and atmospheric oxidation chemistry, during a four-month controlled drought and rewetting experiment[7]. Surprisingly, the emitted enantiomers showed distinct diel emission peaks, which responded differently to progressive drying. Isotopic labelling established that vegetation emitted mainly de novo-synthesized (−)-α-pinene, whereas (+)-α-pinene was emitted from storage pools. As drought progressed, the source of (−)-α-pinene emissions shifted to storage pools, favouring cloud formation. Pre-drought mixing ratios of both α-pinene enantiomers correlated better with other monoterpenes than with each other, indicating different enzymatic controls. These results show that enantiomeric distribution is key to understanding the underlying processes driving monoterpene emissions from forest ecosystems and predicting atmospheric feedbacks in response to climate change.

So far, little attention has been given to the different chiral forms of monoterpenes ((+) and (−)), as both enantiomers have identical physical properties and rates of reaction with OH and $O_3$ (ref. [8]), therefore, most atmospheric field and modelling studies do not differentiate them[4–6]. However, this implicitly assumes that the sources and sinks of both enantiomers are identical, even though the individual enantiomer production pathways and drivers are uncertain. Recent forest measurements showed unequal (non-racemic) concentrations of enantiomers that sometimes do not even correlate with each other[9,10], indicating distinct source mechanisms. Although some reports suggest that the biosynthesis of enantiomers is homogenous throughout an individual plant[10], leaves, bark and soil litter within a homogenous forest have distinct chiral signatures[11], strongly suggesting that the emission and removal processes of these chiral species (and, hence, monoterpenes generally) are not adequately understood.

Isoprene emission is better understood than monoterpene emission, with generally precise model-prediction and measurement agreement[12,13]. Isoprene synthesis occurs by the 2-C-methyl-D-erythritol 4-phosphate pathway, in which photosynthetically assimilated $CO_2$ is converted to the isoprene precursor, isopentenyl diphosphate, and directly emitted from the leaf (de novo emission)[12]. Monoterpene synthesis also occurs by the 2-C-methyl-D-erythritol 4-phosphate pathway, but some monoterpenes are synthesized by the mevalonate pathway. Both pathways result in the production of isopentenyl diphosphate, which combines with its isomer, dimethylallyl diphosphate, to form the common monoterpene precursor geranyl diphosphate[14,15]. Enzymes known as terpene synthases transform geranyl diphosphate into an array of monoterpenes, such that chiral monoterpenes produced by a particular enzyme are typically in one chiral form, (−) or (+)[16,17]. Monoterpenes can be emitted by de novo emission or released from storage pools, thus, decoupled from time of biosynthesis. Broad leaf plant species typical of the tropics usually store monoterpenes non-specifically throughout the leaves, mainly in the lipid phase but also a small amount in the aqueous phase within the leaf[18,19]. The processes regulating monoterpene production and potential storage probably determine the overall chiral emission signature of the plant, and it is unclear how these will change in response to extreme climate events such as drought. Droughts are expected to become more frequent throughout the twenty-first century[20], causing disruptions to the functioning of ecosystems[21] and emissions of volatile organic

[1]Atmospheric Chemistry, Max Planck Institute for Chemistry, Mainz, Germany. [2]Ecosystem Physiology, Faculty of Environment and Natural Resources, Albert-Ludwig-University of Freiburg, Freiburg, Germany. [3]School of Chemistry, The University of Edinburgh, Edinburgh, UK. [4]UK Centre for Ecology & Hydrology, Edinburgh, UK. [5]Biosphere 2, University of Arizona, Oracle, AZ, USA. [6]Honors College, University of Arizona, Tucson, AZ, USA. [7]Department of Environmental Sciences, University of Basel, Basel, Switzerland. [8]School of Natural Resources and the Environment, University of Arizona, Tucson, AZ, USA. [9]Climate and Atmosphere Research Center, The Cyprus Institute, Nicosia, Cyprus. ✉e-mail: jonathan.williams@mpic.de

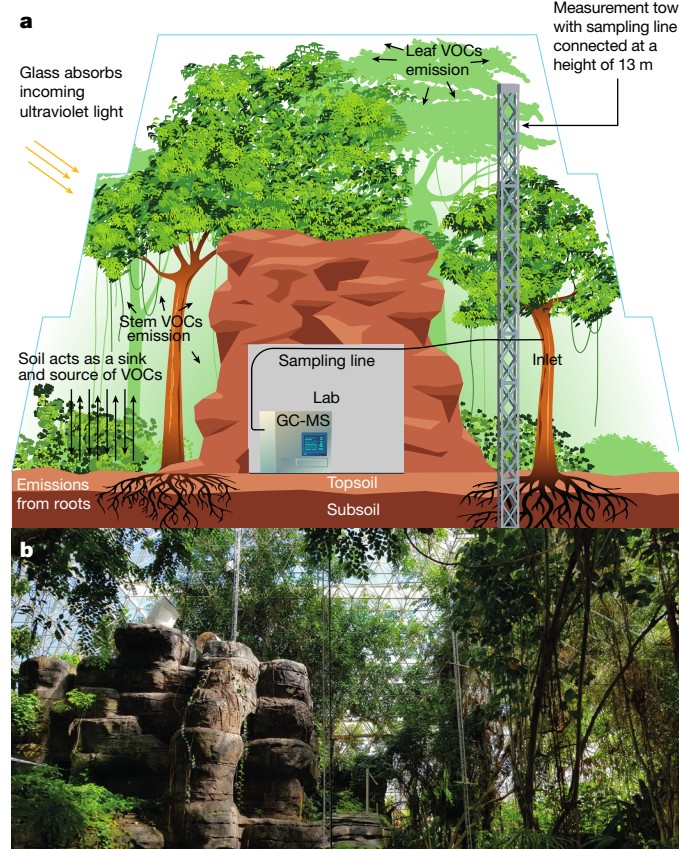

Fig. 1 | **Inside the Biosphere 2 Tropical Rain Forest. a**, Schematic of the Biosphere 2 Tropical Rain Forest biome. **b**, Photograph taken inside the biome (photo J. Byron).

compounds (VOCs) from forests[22]. Reported monoterpene emission responses to drought are highly variable and dependent on the individual plant, making empirically based achiral emission inventories unfaithful[23–27]. However, because chiral compounds link directly to the underlying enzymatically driven processes, they could form the basis of an improved emission scheme.

We separated and measured the enantiomers for α-pinene, camphene and limonene, as well as (−)-β-pinene, γ-terpinene and isoprene, at hourly intervals over almost four months within the enclosed Biosphere 2 Tropical Rain Forest (B2-TRF) by online gas chromatography–mass spectrometry (GC-MS) (Fig. 1). Trans-β-ocimene and β-myrcene were also detected but could not be resolved using the method for the online GC-MS. The B2-TRF was continually flushed with outside air and all incident light wavelengths below 385 nm were filtered by the internal mylar sheet within the surrounding glass panels[28]. After three weeks of ambient measurements to establish the normal pre-drought condition (day of year (doy) 252–280), a 9.5-week drought was imposed to cause physiological change and drive differential responses in the biochemical processes (doy 281–337). The drought stage developed two phases (mild and severe drought), with respect to two relative humidity (RH) minima, soil moisture decline and vegetation response. At the end of the drought, water was added to the deep soil for three days (doy 337–340), followed by 'rain' delivered from overhead sprinklers (doy 347–356). Isotopically labelled $^{13}CO_2$ was added twice to the enclosed atmosphere during pre-drought and severe drought, enabling differentiation between de novo and storage pool emissions using an offline gas chromatography–isotope-ratio mass spectrometer (GC-IRMS). $^{13}C$ labelling combined with long-term atmospheric flux monitoring allowed us to precisely determine how drought affects fluxes and sources of distinct monoterpene enantiomers.

## Distinct trends of enantiomers

Total monoterpenes (consisting of (−)-α-pinene, (+)-α-pinene, (−)-β-pinene, (−)-limonene, (+)-limonene, (−)-camphene, (+)-camphene and γ-terpinene) stayed relatively constant during pre-drought but peaked after 23 days (in early drought) and again, more strongly, after 56 days (in severe drought) (Fig. 2a). During deep-water rewet, when water was reintroduced to the lowest soil levels, the total monoterpene concentrations started to decrease. This decrease continued after rain (rain rewet) but did not fully recover to pre-drought levels by the end of the measurement period. The same pattern of two concentration peaks (corresponding to early and severe drought periods) was also observed when the enantiomeric monoterpenes were separated; however, the enantiomer peak sizes showed strongly contrasting behaviour. Daytime concentrations of (−)-α-pinene were higher in the early drought, whereas for (−)-β-pinene, the severe drought concentrations were ten times that of the early drought (Fig. 2b). The (+)-α-pinene peak concentrations were approximately equal in both early and severe stress conditions. For night-time concentrations, (−)-α-pinene, (+)-α-pinene and (−)-β-pinene all showed higher concentrations during severe drought (Fig. 2c). At the end of the severe drought when emissions of the chiral monoterpenes begin to decrease, the stress marker hexanal was observed to increase, indicating leaf damage[7]. After rewetting by rain, night-time values of (−)-β-pinene, (−)-α-pinene and (+)-α-pinene all returned to pre-drought levels (Fig. 2c). Although daytime concentrations of the enantiomers also all decreased from their severe drought maxima, they did not reach pre-drought levels. During pre-drought, the (−)-α-pinene concentration correlated better with the concentration of (−)-β-pinene, (−)-limonene, (+)-limonene and (+)-camphene than it did with the concentration of (+)-α-pinene (Extended Data Fig. 1a,b). Inversely, during severe drought, the concentration of (−)-α-pinene correlated better with (+)-α-pinene than with any other measured compound. Furthermore, during night-time, the enantiomers correlated well, whereas during daytime, they exhibited independent patterns (when de novo emissions were important) (Extended Data Fig. 1c,d). Notably, the current monoterpene emission model-based expectation that drought would elicit equivalent responses in (−)-α-pinene and (+)-α-pinene was not true.

Although (−)-α-pinene consistently dominated the total monoterpene emissions, (−)-β-pinene overtook (+)-α-pinene to become the second most abundant monoterpene during severe drought (Fig. 2e). Thus, the ratio of (+)-α-pinene to (−)-β-pinene could be used as a proxy of drought severity in this experiment. It should be noted that fluxes of monoterpenes from the soil did not affect these enantiomeric ratios, as samples taken periodically throughout the experiment showed that the soil maintained a modest steady uptake of enantiomeric monoterpenes throughout (Extended Data Fig. 2a). Therefore, soil uptake did not drive the enantiomeric fractionation observed. Furthermore, the air was strongly mixed with fans, resulting in the measurement of the total ecosystem response rather than a single species. The periodic measurements of monoterpene emissions from four *Clitoria fairchildiana* trees and four *Piper* sp. plants (from cuvettes) are provided in Extended Data Fig. 2b,c. The individual plant responses were amalgamated in atmospheric measurements, showing that the atmospheric measurements were the net response of the ecosystem. Furthermore, the fluxes of isoprene and monoterpenes relative to the land surface area and tree biomass carbon were calculated to aid comparison with the real world (Extended Data Table 1).

These responses in monoterpenes contrast those of isoprene dynamics (measured by GC-MS; see Methods for details). During pre-drought, isoprene in the tree canopies increased by a factor of 3 over 26 days, reaching average concentrations of about 300 ppb (Fig. 2a), probably because the topsoil moisture (5 cm soil moisture) decreased from 35% to 26% and the strong soil uptake of isoprene weakened before the drought started[29,30]. During early drought, isoprene and total

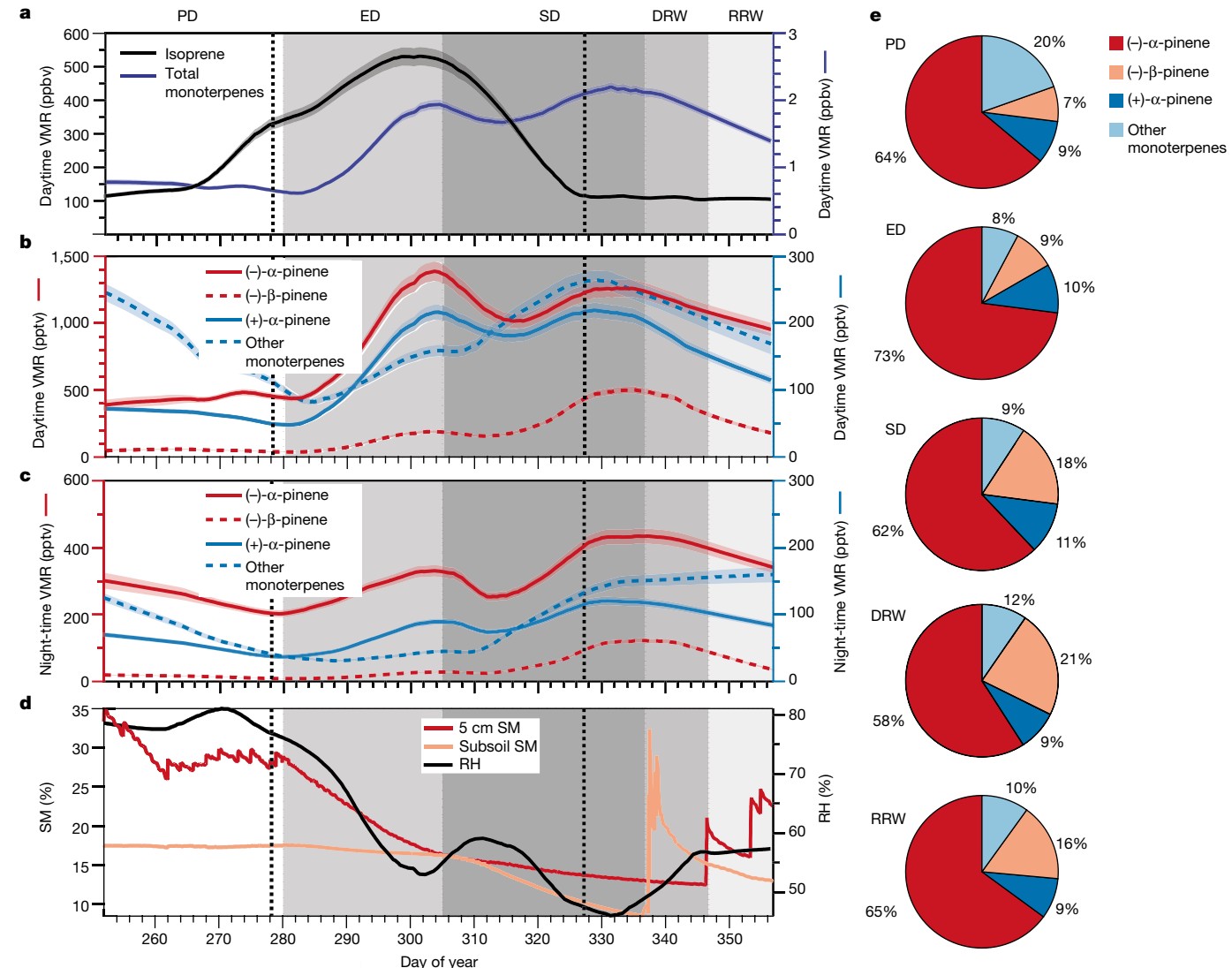

**Fig. 2 | Long-term trends are different between monoterpene enantiomers, especially during daylight hours.** Monoterpene and isoprene data are divided into five stages, indicated by the bands: pre-drought (PD), early drought (ED), severe drought (SD), deep-water rewet (DRW) and rain rewet (RRW). The timing of the $^{13}CO_2$ pulses is indicated by the dotted black lines. **a**, Daytime isoprene and total monoterpene volume mixing ratios (VMR). The shaded region around the lines represents the absolute measurement uncertainty. **b**, Average daytime VMR for (−)-α-pinene and (+)-α-pinene and other monoterpenes. **c**, Average night-time VMR for (−)-α-pinene and (+)-α-pinene and other monoterpenes. For **b** and **c**, the shaded region around the lines represents the calculated measurement uncertainty. **d**, Soil moisture (SM) and relative humidity (RH). Note the different scales for enantiomers. **e**, Pie charts showing the daytime composition of the enantiomeric monoterpenes during each stage. Other monoterpenes includes (−)-camphene, (+)-camphene, (−)-limonene, (+)-limonene and γ-terpinene.

monoterpenes increased in parallel, with isoprene peaking earlier. By the severe drought period, the topsoil moisture had decreased from about 26% to 15% and average isoprene concentrations decreased and plateaued at around 100 ppb, equivalent to initial pre-drought values, whereas the total monoterpenes continued to increase again in severe drought. Thus, under severe drought, the monoterpene to isoprene ratio was notably higher than during early drought. No substantial OH oxidation chemistry can occur in the B2-TRF because glass does not transmit light at wavelengths that generate OH, and ozone within fresh incoming air is lost to the surfaces of large air-handling units. The absence of any important photochemistry is reflected in the ratio of isoprene to its oxidation products (which is 100 times richer in isoprene than in typical Amazon rainforest measurements[31]) and in the isoprene to monoterpene ratio (which favours isoprene in the Biosphere 2 by a factor of about three, owing to its faster OH reaction coefficient). Ozone was also measured post-campaign within the B2-TRF and found to be at very low concentrations, on average, 1.1 ± 0.7 ppb, whereas outside the Biosphere 2, the air was found to contain an average ozone concentration of 49.2 ± 1.2 ppb.

## Distinct enantiomer emission sources

On two days during the experiment (pre-drought and severe drought), $CO_2$ labelled with the heavy $^{13}C$ isotope ($^{13}CO_2$) was introduced into the B2-TRF atmosphere to distinguish between de novo and storage-type monoterpene emissions (Fig. 3). For (−)-α-pinene, the emissions became more enriched in $^{13}C$ during both pulses. Atmospheric samples taken post-pulse show that, on average, the baseline ε$^{13}C$ values of (−)-α-pinene declined to pre-pulse values. This shows that (−)-α-pinene emissions were predominantly de novo but it should not be completely ruled out that during pre-drought, a small fraction also entered the storage pools from which it was emitted after the labelled $CO_2$ was

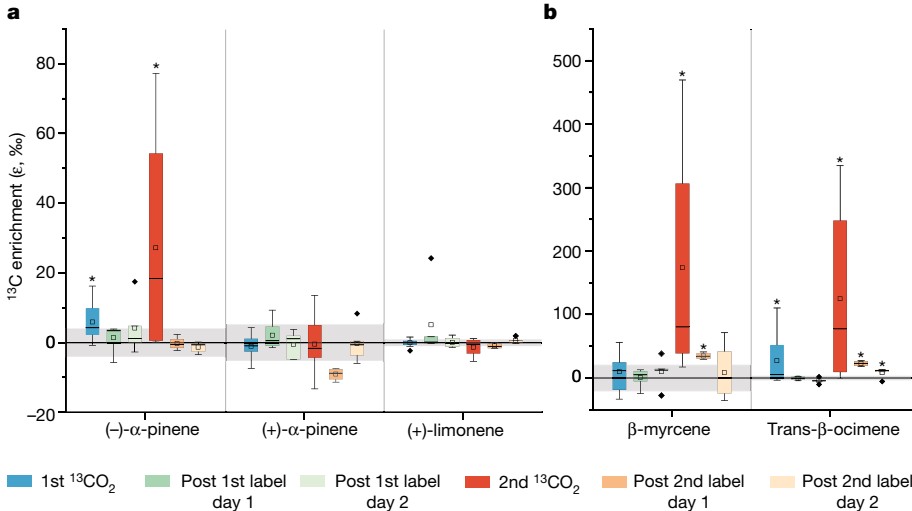

**Fig. 3 | Emissions of α-pinene enantiomers are not equally enriched in ¹³C.** Carbon sources (de novo or storage) for monoterpene emissions were clearly separated by ε¹³C values of monoterpenes and their enantiomers after ¹³C-enriched CO₂ was added during one morning in pre-drought (1st ¹³CO₂) and severe drought (2nd ¹³CO₂) phases. ¹³CO₂ gas was introduced into the atmosphere so that plants taking up CO₂ and directly producing immediate monoterpene emissions (de novo) would produce emissions enriched in ¹³C. Therefore, emissions that did not become enriched in ¹³C came from storage pools. **a**, Enrichment of chiral monoterpenes. **b**, Enrichment of non-chiral monoterpenes. Grey shading represents the standard deviation of the ε¹³C values of the compounds in ambient air when there is no ¹³CO₂ pulse. The black line through the grey boxes represents the mean. The box plots present the median and 25th and 75th percentiles. The small squares represent the mean and the whiskers represent the maximum and minimum acquired data points that are not as considered outliers. Significantly ¹³C-enriched values are indicated by the asterisk (*) above the box (that is, results are significant if $P \leq 0.05$). n values are given in Extended Data Table 3.

flushed from the B2-TRF. By contrast, no notable incorporation of ¹³C was observed for (+)-α-pinene and (+)-limonene, indicating that these enantiomers were generated separately and emitted primarily from storage pools[32]. Therefore, the daytime increases of these monoterpenes during drought (Fig. 2b) came from an increase in storage-pool emissions.

The observed increases in ε¹³C values clearly indicate that (−)-α-pinene, trans-β-ocimene and β-myrcene were synthesized from freshly assimilated photosynthetic carbon and produced during the labelling pulse. The incorporation of ¹³C by (−)-α-pinene, trans-β-ocimene and β-myrcene increased during the second pulse, even though overall ¹³C assimilation declined under drought, probably because more freshly assimilated carbon is used for the production of these specific compounds. Trans-β-ocimene and β-myrcene are important emissions because they react quicker than other monoterpenes with harmful reactive oxygen species. This agrees with measurements from the Amazon rainforest, in which emissions of ocimene and β-myrcene increased from heat-stressed leaves[5].

## Distinct diel cycles of enantiomers

Two distinct types of diel cycle were observed during pre-drought. One, followed by (−)-α-pinene and (−)-β-pinene, was aligned to photosynthetically active radiation (PAR) and assimilation rate (A) peaking earlier at noon, whereas the second, followed by (+)-α-pinene and (+)-limonene, was aligned to vapour pressure deficit (VPD) and temperature peaking in the early afternoon. Whole-day and daytime average values of VPD and temperature for the understory and canopy are provided in Extended Data Table 2. Current atmospheric models predict α-pinene emissions as a function of temperature and light and, therefore, would erroneously place peak monoterpene emission midway between the real peaks and be unable to reproduce the drought-induced changes revealed by resolving enantiomers[33].

During pre-drought and early drought, (−)-α-pinene and (−)-β-pinene peaked with maximum A between 11:00 and 12:00. By contrast,

(+)-α-pinene and (+)-limonene peaked between 14:00 and 15:00, coincident with maximum temperature and VPD (Fig. 4c). With the transition into severe drought, the maxima of the (−)-α-pinene and (−)-β-pinene diel cycles shifted progressively later in the day, merging with the diel cycles of (+)-α-pinene in the afternoon, while the assimilation rate and PAR (A) declined (less carbon uptake by the vegetation) and VPD increased. With rewetting, the assimilation rate began to recover (increased carbon uptake), concurrent with a shift in peak (−)-α-pinene and (−)-β-pinene from 14:00 back to 12:00. Hence, the shift of the daily maximum of (−)-α-pinene to the afternoon with progressive drying potentially suggests that the emissions are less de novo and more storage pool in character (Fig. 4a). As the diel cycles of (−)-β-pinene followed the same temporal pattern as (−)-α-pinene, it is probable that (−)-β-pinene also transitioned from de novo emission to storage-pool emission. The shift in the timing of the emission is particularly important, as this will affect processes related to the formation of cloud condensation nuclei. Towards the afternoon, there is a shift in the partitioning between evaporation and sensible heat flux that favours the latter. More sensible heat flux enhances turbulence in the afternoon, which would facilitate vertical transport of the later emitted species to cooler, more oxidative regions.

Monoterpenes are generally stored in the lipid phase rather than the aqueous phase within the leaf[18,19]. Monoterpenes are relatively water insoluble and partition rapidly between the aqueous and gas phases, according to their Henry's constant[34]. The aqueous-phase storage is small and empties quickly in the morning when the stomata open and water is lost from the leaf to the atmosphere, therefore, any monoterpene emissions from aqueous-phase storage are probably negligible[18]. Monoterpenes possess large octanol/water partition coefficients (≈20,000–30,000), meaning that they can be stored in relatively large fractions in the lipid phase, from which they are more slowly emitted to the atmosphere[18]. A plausible explanation of the emission behaviour is that, throughout the measurement period, (+)-α-pinene and (+)-limonene were stored in the lipid phase, leaking slowly into the atmosphere, peaking later in the day than the de novo emissions. As

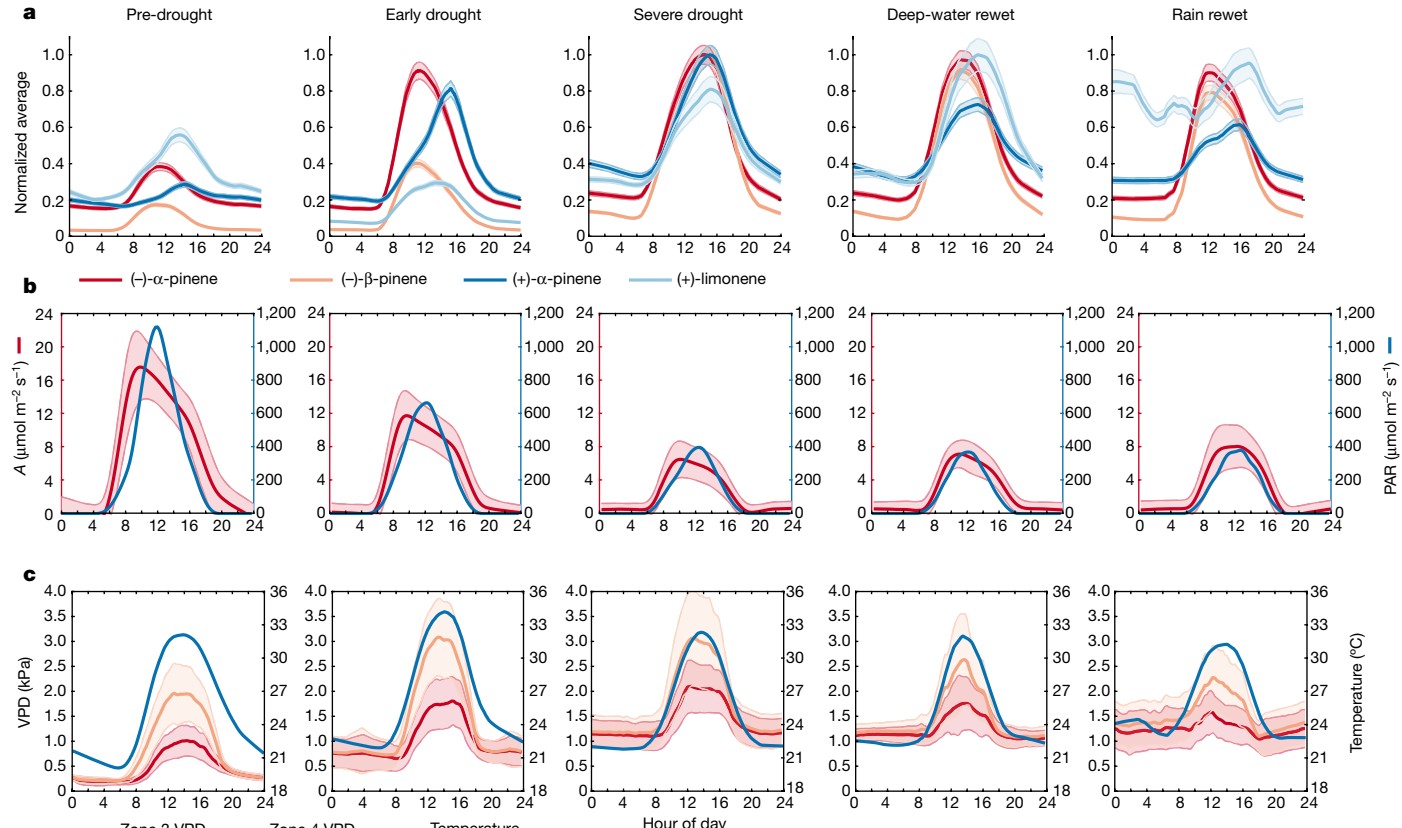

**Fig. 4 | Diel cycles of α-pinene enantiomers become more aligned with increasing drought.** Comparison of diel cycles of selected monoterpenes and their enantiomers with light, temperature and soil moisture across the experiment suggest emission driver changes in some monoterpenes. **a**, Average diel cycles for (−)-α-pinene and (+)-α-pinene, (−)-β-pinene and (+)-limonene. The monoterpene diel cycles were normalized by their respective maxima across all averaged diel bins, which was found to be during severe

drought for both compounds. The shaded region around the lines is the average absolute uncertainty. **b**, Assimilation (*A*) and photosynthetically active radiation (PAR). The shaded region around the assimilation rate line represents 1σ. **c**, Vapour pressure deficit (VPD) and temperature. Zones 3 and 4 were the uppermost sections of the rainforest enclosure (Extended Data Fig. 3a). The shaded region around the VPD lines represents 1σ.

photosynthesis decreased and biomass degradation increased, the emission of (−)-α-pinene and (−)-β-pinene transitioned from mainly de novo synthesis to mainly emission from storage pools.

By grouping enzymes that share similar attributes (that is, light and temperature dependency, principal monoterpene product), a three-enzyme-group–two-reservoir model can explain the observed isoprene and monoterpene emissions (Extended Data Fig. 4). Isoprene is generated by light-activated isoprene synthase (enzyme group 1) and (−)-α-pinene and (−)-β-pinene are generated by light-activated enzyme group 2 and emitted directly (de novo). However, both (−)-α-pinene and (−)-β-pinene are also partitioned to the lipid phase, in which they are released on drought stress. Enzyme group 3 is responsible for synthesizing (+)-α-pinene and (+)-limonene continuously without light activation, which also partition to the lipid phase.

The changes in ecosystem emissions in relation to potential particle formation and growth are important because increased cloud condensation nuclei production efficiency, cloud formation and subsequent rain would represent a possible negative biosphere–atmosphere feedback to the drought. In early drought, de novo monoterpene emissions increase. High fluxes of reactive isoprene will act to suppress OH radical levels above the canopy, allowing emissions to reach higher, cooler altitudes, at which the newly formed aerosols may invigorate convection[35]. In severe drought, isoprene emissions, which do not produce particles or extremely low VOCs efficiently, decrease, whereas monoterpene emissions increase further. Among the monoterpenes, the most prolific increase is of α-pinene, which oxidizes to extremely

low VOCs and particles with high yield. The results presented here are, therefore, consistent with the negative feedback mechanism suggested above. Furthermore, the shift in monoterpene emissions during drought to later in the afternoon means they are released in periods with higher sensible heat flux and turbulence, which would facilitate the transport of monoterpene-rich air parcels to the levels of cloud formation. Extended Data Figure 5 shows the measured enantiomers of α-pinene with the temperature and light-based predictions of the emission model MEGAN. During drought, the measured emissions substantially exceed those predicted and the modelled data do not account for the shift observed in the diel emission profile. The difference in peaks of light and temperature are less important in the real world than within the enclosure, which will probably reduce the error introduced by a pooled representation of the enantiomers. However, the larger difference between the peaks of light and temperature in the Biosphere 2 enabled the discovery of the underlying enantiomeric emission differences.

The emission characteristics of enantiomers were deciphered because the B2-TRF provides a unique venue to conduct a drought under controlled conditions, with comprehensive biological measurements, in addition to atmospheric isotopic labelling in an environment of greatly reduced atmospheric chemistry. These results indicate that the degree of drought stress in a tropical rainforest can be gauged by either the afternoon-to-morning ratios of (−)-α-pinene or by the fractional contribution of (−)-β-pinene to the sum of monoterpenes, which almost tripled from pre-drought to severe drought. Unexpectedly, the

(−) enantiomers from different monoterpenes exhibited the same diel behaviour, whereas opposite enantiomers from the same monoterpenes exhibited different diel cycles. It is remarkable that the de novo and storage-pool sources of monoterpene emissions from an ecosystem, and the carbon cycling changes in response to drought, can be assessed externally from air measurements if individual enantiomers are considered. The enantiomeric results presented here concur with previous studies that have shown that the current approach of relating VOC emissions simply to light and temperature is inadequate for simulating changes associated with drought. Extra levels of complexity, ideally based on real physical processes such as storage pools and enzyme models, may be considered in light of these new results. This underlines that a new, enantiomerically specific approach to emission modelling, based on the enzyme-group model proposed here, would be more accurate and biologically founded than current approaches[36], in particular with regards to a possible feedback between emission composition and particle production efficiency. Whereas monoterpene enantiomers exhibit no difference in physical properties, in oxidation rates by OH or $O_3$, or in uptake rates to typical Amazon forest aerosol samples[8], recent work has shown that dimeric photochemical product combinations do indeed have different hydrophobicities[37]. It should be noted that these conclusions are based on the assumption that the B2-TRF does represent the characteristic drought response of real-world tropical rainforests in the absence of atmospheric chemistry. We conclude that enantiomerically resolved monoterpenes are required to faithfully assess how key tropical regions, such as the Amazon rainforest, emit monoterpenes and respond to the predicted increase in future extreme drought events.

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

## Methods

### Tropical rainforest mesocosm

The tropical rainforest mesocosm at the Biosphere 2 covers ca. 1,950 m$^2$ and is representative of a managed tropical rainforest ecosystem similar to those found in South America. It is contained under a 27,700-m$^3$ glass ziggurat enclosure and is located near Tucson, AZ, USA[29] (Fig. 1). Within the enclosure, there are 95 species of tropical plants, including 23 species of trees and 67 species of understory plants. The most dominant species are *C. fairchildiana*, *Phytolacca dioica*, *Arenga pinnata*, *Ficus benjamina*, *Syngonium podophyllum*, *Piper* sp., *Musa* sp. and *Pachira aquatica*. To recreate real rainforest conditions, such as low light intensity beneath the canopy, larger trees and understory plants (*Musa* sp., *Piper* sp., *Hibiscus rosa-sinensis*) were planted around the edges of the glass enclosure to block out direct sunlight. Most of the biomass (approximately 85%) is in the large (diameter at breast height >15 cm) trees, about 10% in the understory trees and around 5% in the understory herbaceous species (including *Musa* sp., *Alpinia* sp., *Hedychium* sp. and *Zingiber* sp., planted along the walls). *Clitoria fairchildiana* dominates the canopy (around 33%). *Pterocarpus indicus* and *H. tiliaceus* each take up about 15% and 10% of the canopy, respectively. All other tree species take up 5% or less each.

Further lighting was not used in the tropical rainforest. The soil profile consists of an up to 4 m deep subsoil layer and a topsoil layer of variable thickness (70–100 cm)[30]. Located at the centre of the rainforest is a small artificial 'mountain', within which a laboratory was constructed to house the online GC-MS and where the ambient temperature and humidity inside could be controlled. The Biosphere 2 is, therefore, an ideal data source and test bed for future emission models of tropical rainforests and similar facilities for other forest ecosystems are necessary.

### Drought experiment

This experiment was conducted during the Water, Atmosphere, and Life Dynamics (WALD) campaign. To ensure timely progress of drought, manipulation of the ecosystem moisture began with turning off the aesthetic water features (waterfall, pond, stream) before the experiment on 31 May 2019. Before the start of the drought period, the tropical rainforest was wetted from above with a sprinkler system to simulate rainfall, using approximately 20,000 l of water three times a week. After watering on 7 October 2019, the rainforest biome was left to dry. RH was actively reduced using a large air handler unit during the severe drought (1 November 2019–2 December 2019). Air was first cooled to create condensation and then reheated to maintain temperature. To enhance drought conditions, a persistent water table in the isolated drainage basin of the várzea subhabitat was drained throughout the severe drought period. During some stages, air handler units were used for the removal of humidity by condensation, otherwise, the rainforest biome was left to dry naturally, until 3 December 2019. The first water to the rainforest was introduced at the bottom through a network of drainage pipes under the soil on top of the concrete and steel structure underlying the Biosphere 2. The rainforest was again watered from above using the sprinkler system and about 35,000 l of water at 11:00 on 12 December 2019 and about 36,000 l of water at 11:00 on 19 December 2019. The rainforest was then watered at 00:00 every two days afterwards, adding 20,000 l of water to the rainforest over a 4.5-h period. The rainforest temperature was controlled throughout the experiment and the temperature at 13 m was, on average, between 28 and 32 °C during daylight hours and between 21 and 24 °C during night-time hours[7].

The PAR, temperature and RH were recorded every 15 min using sensors connected to a data logger (PAR sensors (Apogee SQ-110, Campbell Scientific, Logan, UT, USA), temperature and RH with Vaisala HMP45C sensors (Vaisala Oyj, Vantaa, Finland; purchased through Campbell Scientific)). Sensors reported every 15 min to a centralized CR1000 data logger (Campbell Scientific, Logan, UT, USA) with an AM16/32B multiplexer (Campbell Scientific, Logan, UT, USA). The data loggers were connected to a centralized database with NL100 communications modules (Campbell Scientific, Logan, UT, USA). The PAR sensor was located at a height of 13 m on the central measurement tower and the humidity and temperature sensors were located at a height of 13 m on the north-eastern measurement tower, together with the sampling inlet. The soil moisture data presented in Fig. 2d is an average of measurements that were recorded every 15 min from four different soil pits (soil moisture and temperature sensors (SMT-100, Truebner GmbH, Neustadt, Germany) and water potential sensors (TEROS 21, METER Group, Pullman, WA, USA) in all four pits at 5 cm depth and at the soil–concrete interface (subsoil bottom). As the sensors are 30 mm wide and inserted vertically into the soil with the soil depth indicated at the midpoint, each depth is ±1.5 cm.

The rainforest enclosure acted as a semi-enclosed system in which there was constant air exchange with the outside environment. The air-exchange rate from the tropical rainforest enclosure was measured using sulfur hexafluoride ($SF_6$) at low ppb levels as a tracer gas, as it is completely anthropogenic and its concentration is <10 ppt in background air and, therefore, it is only affected by leakage and flushing (Extended Data Fig. 3b). To measure the air-exchange rate, 25 to 30 ml $SF_6$ was injected into the rainforest, thereby generating a concentration of about 1 ppb (around 125 times background air at about 8 ppt). $SF_6$ sampling took place next to the instrument laboratory using a single, filtered inlet connected to ¼" OD Teflon tubing. The concentration of $SF_6$ was measured using a SRI Greenhouse Gas GC (SRI Instruments, Torrance, CA, USA) with an automated sample loop of 1 ml using an electron capture detector at 350 °C. A HayeSep D column at 65 °C was used to separate the $SF_6$ in the sample and the ultrahigh-purity $N_2$ carrier stream from $N_2O$. Samples were collected and analysed every 2.5 min. The exponential decay of the $SF_6$ concentration in the Biosphere 2 rainforest was used to calculate the exchange rate and was reported as % per hour[7,29]. Once the percentage exchange rate of $SF_6$ was obtained and interpolated, the measured data were corrected by using the equation $VMR_c = VMR_u + (VMR_u \times ER)$, in which $VMR_u$ is the uncorrected data, ER is the exchange rate percentage and $VMR_c$ is the corrected data, and incoming VOCs VMRs were assumed negligible. Post campaign, ozone in the B2-TRF was quantified using an ozone analyser (Model 205, 2B Technologies, Boulder, CO, USA).

### Determination of isoprene and monoterpene ambient mixing ratios

From 9 September to 23 December 2019, the ambient air from a height of 13 high within the Biosphere 2 rainforest was continuously drawn at a flow of approximately 800 ml min$^{-1}$ through a main Teflon inlet line, which consisted of 37 m of 0.625 cm (¼") Teflon tubing. 13 m was chosen as the sampling height as this was the height in the enclosure that had the greatest leaf area index. The main inlet line was fitted with a Cole-Parmer EW-02915-31 filter. After approximately 26 m, a T-piece was connected to the main inlet line, which was connected to a thermal desorption unit (TD) (TT24-7xr, Markes International Ltd., UK) using 7 m of 0.3175 cm (1/8") Teflon tubing. All sampling lines were insulated and heated to 50 °C to avoid water condensation within the lines. The line to the TD was continuously purged to avoid the sampling of a dead volume, with a pump situated behind the TD in the flow path. During sampling, this pump drew air from the main inlet line at flows ranging between 70 and 200 ml min$^{-1}$ for 10 min. The collected air was first sampled through a water condenser (Kori-xr, Markes International Ltd., UK). This allowed for the removal of water whilst leaving the target VOCs unchanged. The dehumidified sample was then pre-concentrated onto a cold injection trap at 30 °C (Material emissions, Markes International Ltd., UK). After sampling, the injection trap was purged for 1 min with helium at a flow of 50 ml min$^{-1}$ before being rapidly heated to 300 °C and desorbed for 3 min. The sample was removed from the cold trap

with a helium flow of 3 ml min$^{-1}$, including a split flow of 2 ml min$^{-1}$, and injected into the separating column.

The rainforest ambient air was analysed using a gas chromatograph (6890A, Agilent Technologies, UK). The carrier gas used was research 6.0 grade helium (Airgas, USA) Separation of the sampled compounds was achieved using a 30-m β-DEX 120 column (Sigma-Aldrich GmbH, Germany) with 0.25 mm internal diameter and a 0.25 μm film thickness. The temperature programme used was as follows: 40 °C for 5 min then 40 °C to 150 °C at 4 °C min$^{-1}$ and 150 °C to 200 °C at 30 °C min$^{-1}$. The column flow was set to 1 ml min$^{-1}$.

The gas chromatograph was coupled with a quadrupole mass spectrometer (5973N, Agilent Technologies, UK), operated in selected-ion mode for the identification of mass ions 68, 69, 93, 94, 119, 120, 136 and 137, each with a dwell time of 60 ms.

The identification of the target compounds was achieved by first operating the mass spectrometer in scan mode to obtain full mass spectra to be able to compare with the NIST 70-eV electron ionization library. For further confirmation, a gas standard mixture (Apel-Riemer Environmental Inc., 2019) containing the target compounds was injected into the GC-MS system. The same gas standard mixture was also injected onto sorbent cartridges and subsequently desorbed into a gas chromatography–time-of-flight mass spectrometer operated with identical conditions to the online GC-MS. Using liquid standards, the headspace of the individual compounds was taken onto sorbent cartridges and also desorbed into the gas chromatography–time-of-flight mass spectrometer. The retention times from the chromatograms of the individual compounds were then cross-checked with the chromatogram of the gas standard mixture.

The mass spectrometer was tuned on a weekly basis and the linearity was checked throughout the campaign. The gas standard mixture was injected into the system after each tuning and after ten samples were analysed. Routine calibrations were performed by initially flushing the TD system with the gas standard mixture at a flow of 20 ml min$^{-1}$ for 2 min to remove the dead volume. The calibration gas was then injected with a flow of 20 ml min$^{-1}$ for 5 min directly onto the cold injection trap within the TD. The calibration gas sample was then treated with the same TD GC-MS parameters as the routine sampling. This step was repeated three times before sampling continued. The mass spectrometer responses to the injected calibration gas samples were then plotted against the time since the last mass spectrometer tune to track the mass spectrometer sensitivity drop, which allowed for the correction and calibration of the raw data. To check the linearity, the same procedure was used with the calibration gas injection time being increased in stepwise intervals of 2.5 min from 0.5 to 12.5 min.

Werner et al. described the overall ecosystem response to drought and presented data from a proton-transfer-reaction time-of-flight mass spectrometer (PTR-TOF-MS) to represent isoprene for consistency with the accompanying soil and cuvette flux data[7]. It should be noted that the isoprene data presented in Fig. 1e of Werner et al. are daily averages (24 h), whereas those shown here in Fig. 2a are daytime averages (PAR > 0.1 μmol m$^{-2}$ s$^{-1}$). Here the focus is on enantiomeric monoterpenes, which can only be measured by GC-MS techniques, as pre-separation is required. Therefore, again for consistency, we use the isoprene measured by the same GC-MS instrument. Although broadly similar in the temporal behaviour, the isoprene traces from both systems diverged in concentration during the early drought period (PTR-TOF-MS was lower). Despite rigorous investigation of both systems, no cause for the discrepancy could be found, even with the inlets being closely located to each other. Therefore, we concluded that the only remaining plausible cause for the discrepancy is that the sampling lines for the two instruments had differing flow rates, which sampled different locally influenced air. As the temporal behaviour of isoprene is used only as an indicator of the general behaviour of the de novo emission signal, the short-term differences in isoprene concentrations between the instruments are not important in this context, and the same conclusions can be drawn using the other dataset.

## Data management

The highest individual VMRs measured were during early drought and were in excess of 3 ppb for (−)-α-pinene and 400 ppt for (+)-α-pinene (Extended Data Fig. 1c). However, to evaluate the general trends of all measured compounds through each stage, the hourly total monoterpene and isoprene data were smoothed by applying a Savitzky–Golay filter to retain long-term trends whilst removing short-term fluctuations (Extended Data Fig. 6). The dataset was further split into daytime and night-time, using PAR data collected at the point of measurement. The smooth function (MATLAB) was then used to suppress noise in the trend line for each compound and the uncertainties were propagated using the same functions. To obtain average diel cycles for each compound, the average composition of all data in each of the five stages of the campaign were taken. A moving median calculation with a window length of five data points was applied to each group. The diel cycles were averaged over 435, 526, 349, 136 and 193 data points for pre-drought, early drought, severe drought, deep-water rewet and rain rewet, respectively. β-Myrcene, (+)-β-pinene, α-terpinene and terpinolene were also observed but not included, as they amounted to an average of less than 5% of the average total monoterpene for the entire measurement period. Ocimene was also observed but not calibrated with the online GC-MS system.

The VPD (in kPa) was calculated from the temperature ($T$) and RH measurements according to ref. [38], VPD = 0.6108(1 − RH/100)$e^{17.27T/(237.3+T)}$. Zones 3 and 4 are the two height zones that contain most of the canopy in the Biosphere 2 rainforest (Extended Data Fig. 3a). The environmental conditions were averaged over all the sensors that were located in these zones. The net ecosystem exchange (NEE) was calculated every 15 min based on the change in moles of $CO_2$ in the rainforest ecosystem and the amount of $CO_2$ lost or gained with the air exchange with the outside, NEE = ((($CO_2^t$ − $CO_2^{t-1}$) + ($CO_2^{27m}$ − $CO_2^{Outside}$) × ER))/Area, in which $CO_2^t$, $CO_2^{t-1}$, $CO_2^{27m}$ and $CO_2^{Outside}$ are the moles of $CO_2$ at the time calculated, previous time step, 27 m or top of the rainforest where the air flows out, and the outside air coming into the rainforest. Area stands for the soil surface area of the rainforest. The moles of $CO_2$ were calculated on the basis of the ideal gas law and the $CO_2$ concentration, moles of $CO_2$ = ($V$ × $P$)/((273.15 + $T_{Ave}$) × $R$) × [$CO_2$]/10$^6$, in which $V$ denotes the representative volume for the $CO_2$ measurement (either the rainforest volume fraction or the volume of air exchanged), $P$ denotes the pressure (measured inside and outside the Biosphere 2 rainforest using WeatherHawk WXT530, Vaisala Oyj, Vantaa, Finland), $T_{Ave}$ denotes the average air temperature for the measurement zone or the outside air temperature (measured using HMP45C temperature and humidity sensors (Vaisala Oyj, Vantaa, Finland), $R$ denotes the gas constant and [$CO_2$] denotes the $CO_2$ concentration measured inside the Biosphere 2 with GMP343 $CO_2$ sensors (Vaisala Oyj, Vantaa, Finland) and outside the Biosphere 2 with GMP220 sensors (Vaisala Oyj, Vantaa, Finland) and inside the air inflow with an Aerodyne Dual QCL (Aerodyne Research Inc., Billerica, MA, USA). The ecosystem assimilation ($A$, μmol m$^{-2}$ s$^{-1}$) was calculated from the NEE and assuming that the night-time respiration ($R$) was representative for the daytime respiration (a reasonable assumption in tropical forest ecosystems[39,40]), $A$ = NEE − $R$.

## $^{13}CO_2$ pulse-labelling experiment

The $^{13}CO_2$ pulses were carried out on 5 October 2019 at 08:00 (MST) and 23 November 2019 at 09:00 (MST) to coincide with peak photosynthetic activity[41]. A deliberate effort was made to proceed with the pulse experiments on days with high amounts of direct sunlight, when there would be a high rate of photosynthesis, to maximize $CO_2$ uptake. During the first pulse, the rainforest was fumigated with 10 lpm of 99% $^{13}CO_2$ (MilliporeSigma, Burlington, MA, USA) for 15 min. To balance reduced carbon assimilation rates during the drought, 20 lpm of 99%

$^{13}CO_2$ was released over 15 min during the second pulse. The $\delta^{13}C$ value of atmospheric $CO_2$ in the rainforest was monitored throughout each pulse using a tunable infrared laser direct absorption spectrometer (Aerodyne Research, Billerica, MA, USA). After 4 h during the first pulse and 5.2 h during the second pulse, the flow of air through the rainforest was increased and, at midday, windows were temporarily removed from the enclosure of the B2-TRF and excess $^{13}CO_2$ was ventilated to the outside air, so that the entry of $^{13}C$ into the mesocosm could be more accurately traced back to a fixed point in time.

### Determining enantiomer $^{13}C$ isotope ratios

Pairs of monoterpene enantiomers abundant in the air were analysed in the tropical rainforest ecosystem at a height of 13 m at the atmosphere tower. 5, 16, 36, 11 and 14 glass cartridges were sampled during pre-pulse, first pulse, first post-pulse, second pulse and second post-pulse, respectively. For terpene accumulation, ambient air was drawn through glass cartridges filled with about 100 mg Tenax (Sigma, Germany) as an adsorbent at a controlled flow rate of 200 ml min$^{-1}$ for 90 min using a handheld pump (SKC Ltd., Dorset, UK). Glass cartridges were kept at 4 °C until analysis. The samples were analysed at the University of Freiburg on a system consisting of a gas chromatograph (7980, Agilent Technologies, Germany) coupled to a mass-selective detector (5975C, Agilent Technologies, Germany) and equipped with a TD (Gerstel, Germany) and a cold injection system (Gerstel, Germany). For the analysis of $^{13}C$ isotope ratios, this system was coupled to an isotope-ratio mass spectrometer (isoprime precisION, Elementar Analysensysteme GmbH, Langenselbold, Germany) by means of a combustion furnace (GC5 interface, Elementar Analysensysteme GmbH, Langenselbold, Germany). For analysis, air-sampling glass cartridges were heated to 220 °C for 5 min to thermodesorb terpenes and channel them into the cold injection system, which was kept at −70 °C. By heating the cold injection system to 240 °C for 3 min, terpenes were directed onto the GC separation column (beta-Dex 120 Chirality, 60 m × 250 µm × 0.25 µm, Supelco, USA) with a helium stream of 1 ml min$^{-1}$. The oven programme started at 45 °C, which was kept for 1 min, and the temperature was then stepwise increased to 60 °C, 150 °C and 210 °C at rates of 2 °C min$^{-1}$, 1 °C min$^{-1}$ and 3.5 °C min$^{-1}$, respectively. The eluate was split and ca. 10% was directed into the mass-selective detector for terpene identification. For this purpose, the mass-selective detector was run in SIM mode detecting $m/z$ 68, 93, 119 and 136. The remainder eluate passed the combustion furnace in which, at a temperature of 850 °C, the terpenes were oxidized to form $CO_2$ and $H_2O$. After elimination of $H_2O$ by a Nafion water trap, the $^{13}C/^{12}C$ ratios of the $CO_2$ were measured by the isotope-ratio mass spectrometer.

Details of isoprene analysis are provided by Werner et al.[7]. Briefly, atmospheric isoprene concentrations were determined at a height of 13 m in the rainforest. For this purpose, ambient air was sucked through ¼" heated perfluoroalkoxy tubing to a proton-transfer-reaction time-of-flight mass spectrometer (4000ultra PTR-TOF-MS, IONICON Analytik, Innsbruck, Austria). Measurements were taken for 5 min, alternating between every 50 and 60 min (labelling 1) and every 30 and 66 min (labelling 2); in these intervals, 2-min averages were used after quality control.

Explicit calibrations with isoprene calibration gas were performed regularly using a dilution curve obtained with a liquid calibration unit (IONICON Analytik, Innsbruck, Austria). Data obtained were processed with the software package PTRwid.

Labelling of isoprene with $^{13}C$ from $^{13}CO_2$ was calculated as the ratio of the abundance of the single $^{13}C$-labelled isoprene isotope ($m/z$ 69 analysed by PTR-TOF-MS as $m/z$ 69) versus total isoprene, that is, the non-labelled isoprene isotope ($m/z$ 68 measured as $m/z$ 69) plus the single labelled isotope. Owing to the natural abundance of $^{13}C$ (1.1%), the background level of the single labelled isoprene was 5.5% considering the five C atoms of the isoprene molecule. This background level was subtracted from the data shown (Extended Data Fig. 7a).

### Statistical information

The $t$-tests used on the data presented in Fig. 3 and Extended Data Fig. 7e were one-tailed, two-sample, unequal-variance $t$-tests that were performed using the MATLAB R2017b software (Extended Data Table 3).

### GC-IRMS data processing

$^{13}C$ isotopologues elute slightly faster from the GC than their $^{12}C$ counterparts, meaning that $\delta^{13}C$ values are not homogenous across the peak[42]. For chromatographically unresolved compounds such as (−)-α-pinene (Extended Data Fig. 7e), integrating from the beginning of the peak to the trough between it and the subsequent unidentified coeluting peak results in $\delta^{13}C$ values that seem artificially enriched in $^{13}C$. Absolute $\delta^{13}C$ values for such compounds cannot be reported. We therefore report relative offsets between measured $\delta^{13}C$ values during the $^{13}CO_2$ pulses and ambient conditions, $\epsilon^{13}C = (({}^{12}C/{}^{13}C)_{pulse}/({}^{12}C/{}^{13}C)_{ambient}) - 1$. Although the relative abundance of (−)-α-pinene to its coeluter changes its apparent $\delta^{13}C$ value, we are confident that these chromatographic effects cannot account for the relative $^{13}C$ enrichment of this compound during the $^{13}CO_2$ pulses for two reasons. First, the $^{13}C$ enrichment of the samples from the $^{13}CO_2$ pulses are substantially enriched relative to the variability in ambient $\delta^{13}C$ values, which are subject to a greater range of relative peak heights. Second, $^{13}C$ enrichment is only apparent during the $^{13}CO_2$ pulses when (−)-α-pinene is integrated in various combinations with the other three peaks in the coeluting hump, but not when any of these other peaks are individually integrated trough-to-trough (Extended Data Fig. 7e).

There is no indication that (+)-α-pinene becomes enriched in $^{13}C$ during either $^{13}CO_2$ pulse. However, given its small size and poor resolution from the preceding unlabelled peak, we cannot definitively rule out slight $^{13}C$ enrichment for (+)-α-pinene. Myrcene, trans-β-ocimene and (+)-limonene all have well-resolved fronts and poorly resolved tails. These compounds show either high $^{13}C$ enrichment during the $^{13}CO_2$ pulses (trans-β-ocimene during both pulses, myrcene during the second pulse) or no enrichment at all (myrcene during the first pulse, (+)-limonene during both pulses). Because these chromatographically similar peaks only show large $^{13}C$ enrichment in some compounds and not others, we consider the apparent presence or absence of label uptake to be robust results for these compounds (Extended Data Fig. 7c,d).

### Soil uptake and emission of monoterpenes experiment

For the investigation of how monoterpenes are taken up and emitted by the soil, three soil chambers made from polyvinyl chloride were placed on pre-installed soil collars around the B2-TRF (Extended Data Fig. 2a). Using sorbent cartridges, samples were taken from the atmosphere, located at the inlet (MR$_{atm}$) of the soil chamber and at the same time from the outlet (MR$_{soil}$). Samples were collected at 200 ml min$^{-1}$ for about 10 min using a handheld pump (SKC Ltd., Dorset, UK). The sorbent cartridges were made from inert coated stainless steel (SilcoNert 2000 (SilcoTek, Germany)). The sorbent consisted of 150 mg of Tenax TA followed by 150 mg of Carbograph 5TD (560 m$^2$ g$^{-1}$). The size of the Carbograph particles was in the range 20–40 mesh. The Carbograph 5TD was supplied by LARA s.r.l. (Rome, Italy) and Buchem B.V. (Apeldoorn, the Netherlands) supplied the Tenax.

## Data availability

All data used in this manuscript are publicly available (https://doi.org/10.5281/zenodo.6517513).

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

**Acknowledgements** We gratefully acknowledge financial support from the European Research Council (ERC consolidator grant #647008 (VOCO$_2$) to C.W.) and from the Philecology Foundation to Biosphere 2 to L.K.M. We would like to thank all members of the B2WALD team for their valuable support, as detailed in the B2WALD contribution list (https://doi.org/10.25422/azu.data.14632662). J.B. was supported by the Max Planck Graduate Center with the Johannes Gutenberg-Universität Mainz (MPGC). This work was supported by the European Commission Horizon 2020 ULTRACHIRAL project (grant no. FETOPEN-737071). We are very grateful to J. Birks from 2B Technologies for providing an ozone instrument for post-campaign tests.

**Author contributions** J.B. had the idea, performed the online GC-MS sampling, analysed the data and wrote the paper. J.K. performed the offline GC-IRMS laboratory analysis and calculated the ε$^{13}$C values. G.P. planned and performed the atmospheric cartridge sampling. J.v.H. calculated the assimilation rate and atmospheric water potential data, provided the meteorological data and helped conduct the drought experiment. S.N.L., L.K.M. and C.W. supervised and planned the project, performed the drought experiment and $^{13}$CO$_2$ labelling experiment, and commented on the paper. J.W. had the idea, supervised the project and wrote the paper. All authors contributed to writing and editing the manuscript.

**Funding** Open access funding provided by Max Planck Society.

**Competing interests** The authors declare no competing interests.

**Additional information**
**Correspondence and requests for materials** should be addressed to Jonathan Williams.

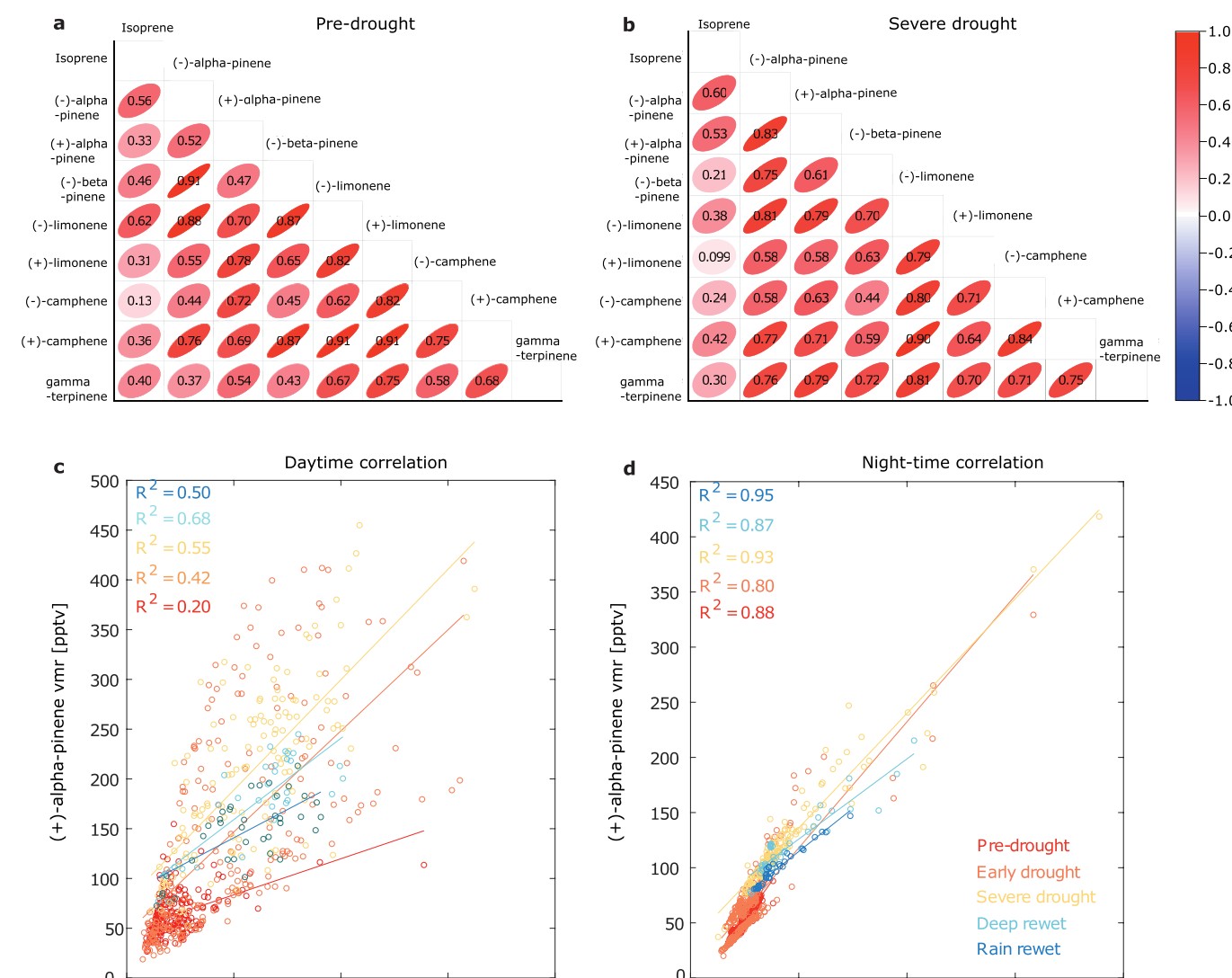

**Extended Data Fig. 1 | Correlations between measured compounds.**
**a**,**b**, Correlation matrix for the measured atmospheric concentrations of monoterpenes. **a**, Pre-drought. **b**, Severe drought. Colours denote strength and direction of the correlation. Correlations are based on Pearson's correlation coefficient. **c**,**d**, Correlations of (+)-α-pinene and (−)-α-pinene during each stage of drought. **c**, Daytime correlation. **d**, Night-time correlation. Data are more correlated during night-time, indicating that the source of the emissions is more similar than the daytime sources.

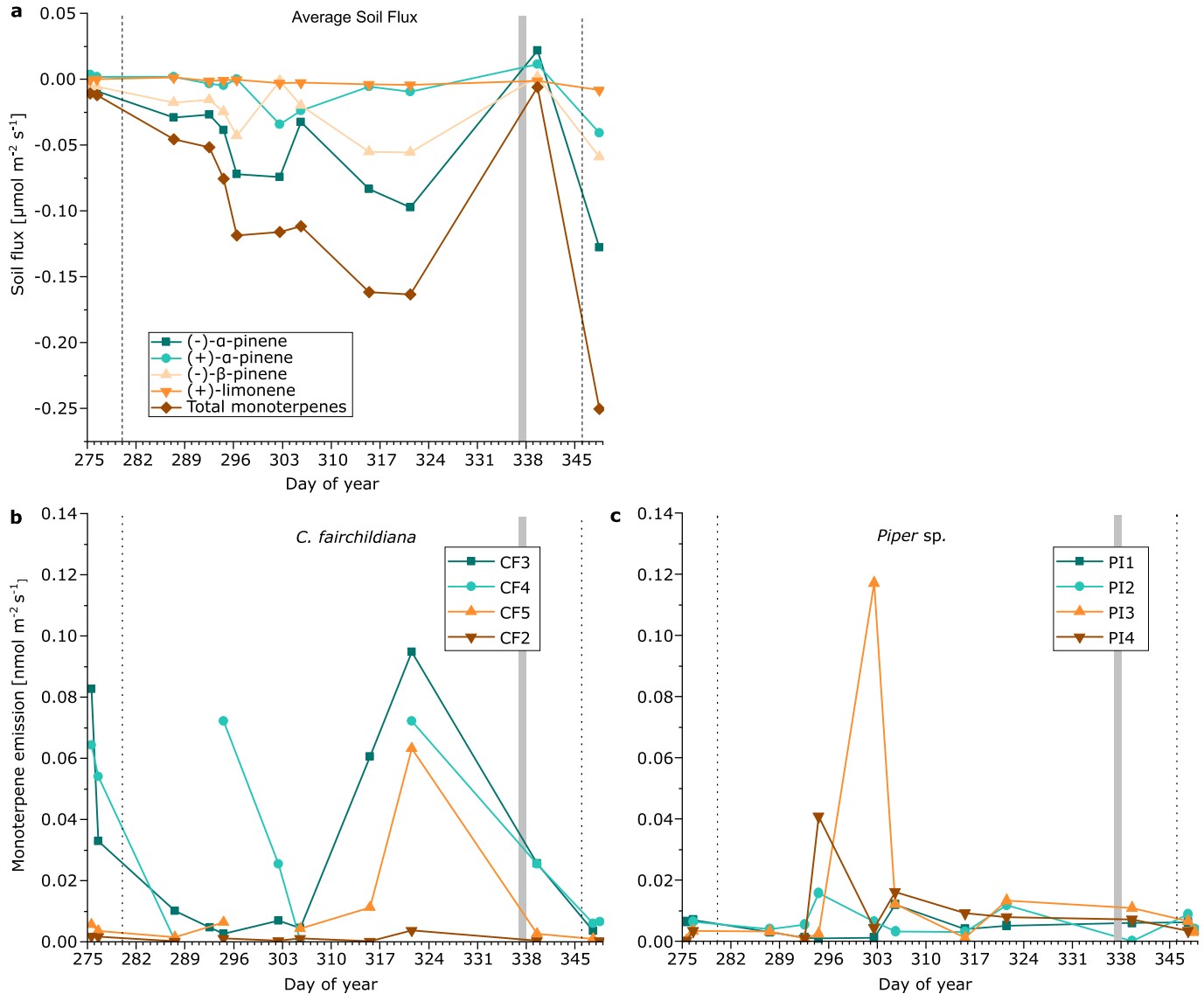

**Extended Data Fig. 2 | Sorbent tube data taken from branch cuvettes and soil chambers. a**, Average soil flux of individual monoterpene species from soil chambers located around the B2-TRF. Values shown are the calculated medians from three soil chamber outlet samples and 2–3 atmospheric sample cartridges taken at the inlet to the soil chamber. **b,c**, Measurements of the emissions from four branch cuvettes on both *C. fairchildiana* (**b**) and *Piper* sp. (**c**) show different emissions trends across the drought period. The dotted line on doy 280 shows the start of the drought and the dotted line on doy 346 shows the end of the drought and the start of the rain rewet. The shaded grey line on doy 337 shows the day of the underground deep rewet.

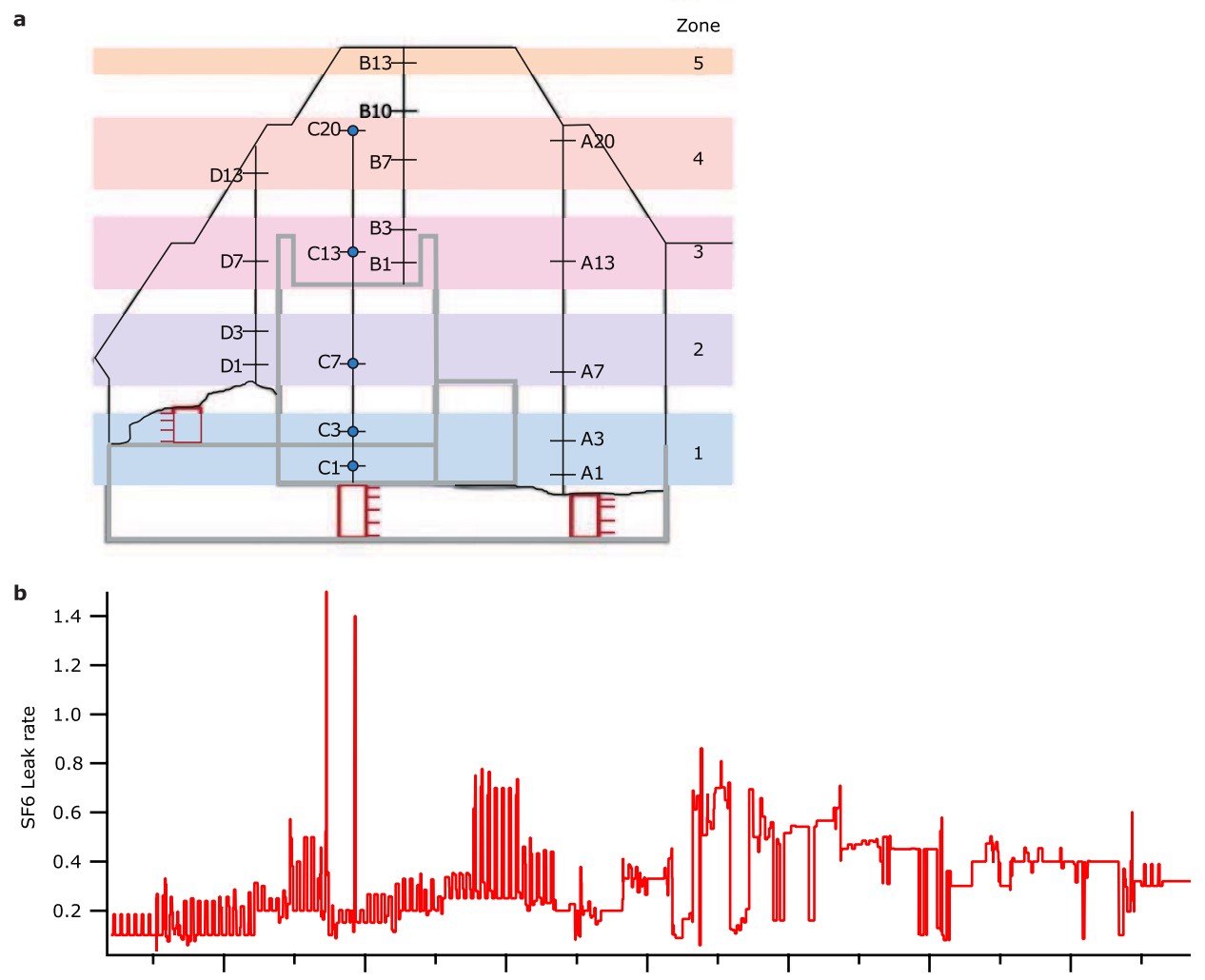

**Extended Data Fig. 3 | Schematic of the B2-TRF microbiome and SF₆ leak rate. a**, The height of the B2-TRF was divided into five zones. A13 is the location where the atmosphere was sampled by GC-MS. **b**, SF₆ leak rate fraction across the entire measurement period. SF₆ was injected into the B2-TRF atmosphere and measured with GC-MS to characterize the air loss rate owing to the flow-through ventilation.

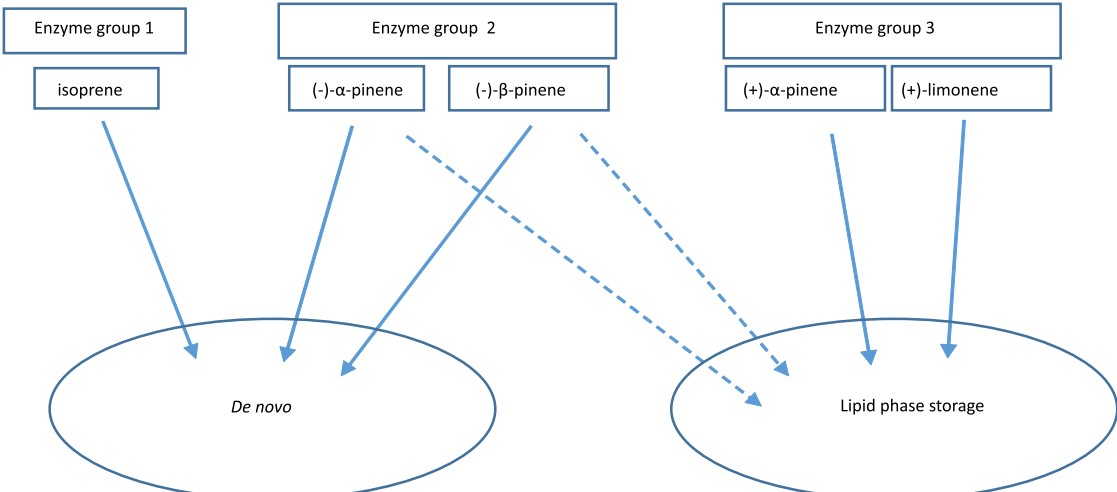

**Extended Data Fig. 4 | Schematic representation of the three-enzyme-group model.** The solid arrows represent the main emission pathway for that compound and the dashed arrows represent a secondary pathway.

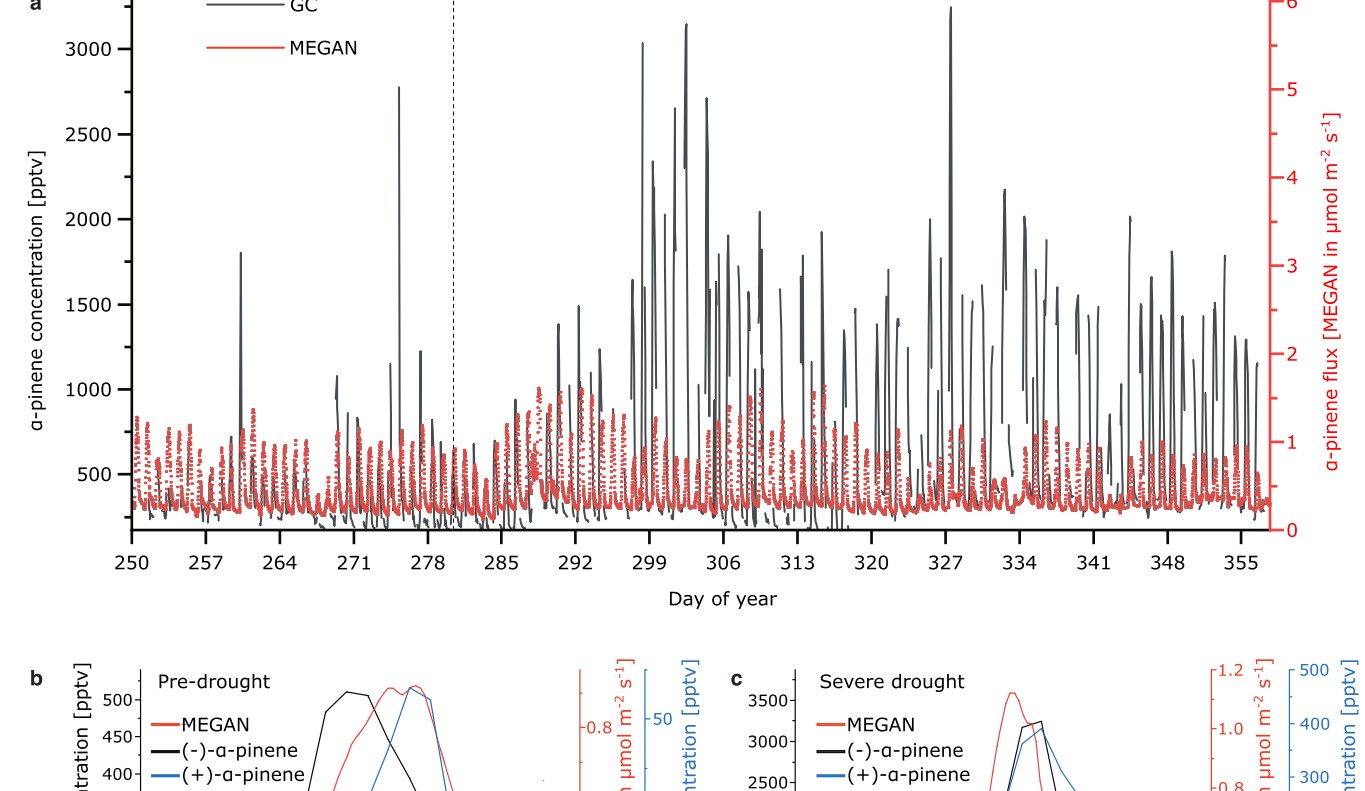

**Extended Data Fig. 5 | Comparison of the measured (−)-α-pinene time profiles with the MEGAN model. a**, The sum of the α-pinene emission and the sum of the α-pinene flux calculated with the MEGAN model as a function of time using the temperature data measured from 13 m on the measurement tower and the PAR data measured outside the Biosphere 2. **b**, Diurnal cycles of the measured concentration of (−)-α-pinene and (+)-α-pinene during pre-drought on doy 280 in addition to the predicted combined emission flux of (−)-α-pinene and (+)-α-pinene. **c**, Diurnal cycles of the measured concentration of (−)-α-pinene and (+)-α-pinene during severe drought on doy 296 in addition to the predicted combined emission flux of (−)-α-pinene and (+)-α-pinene.

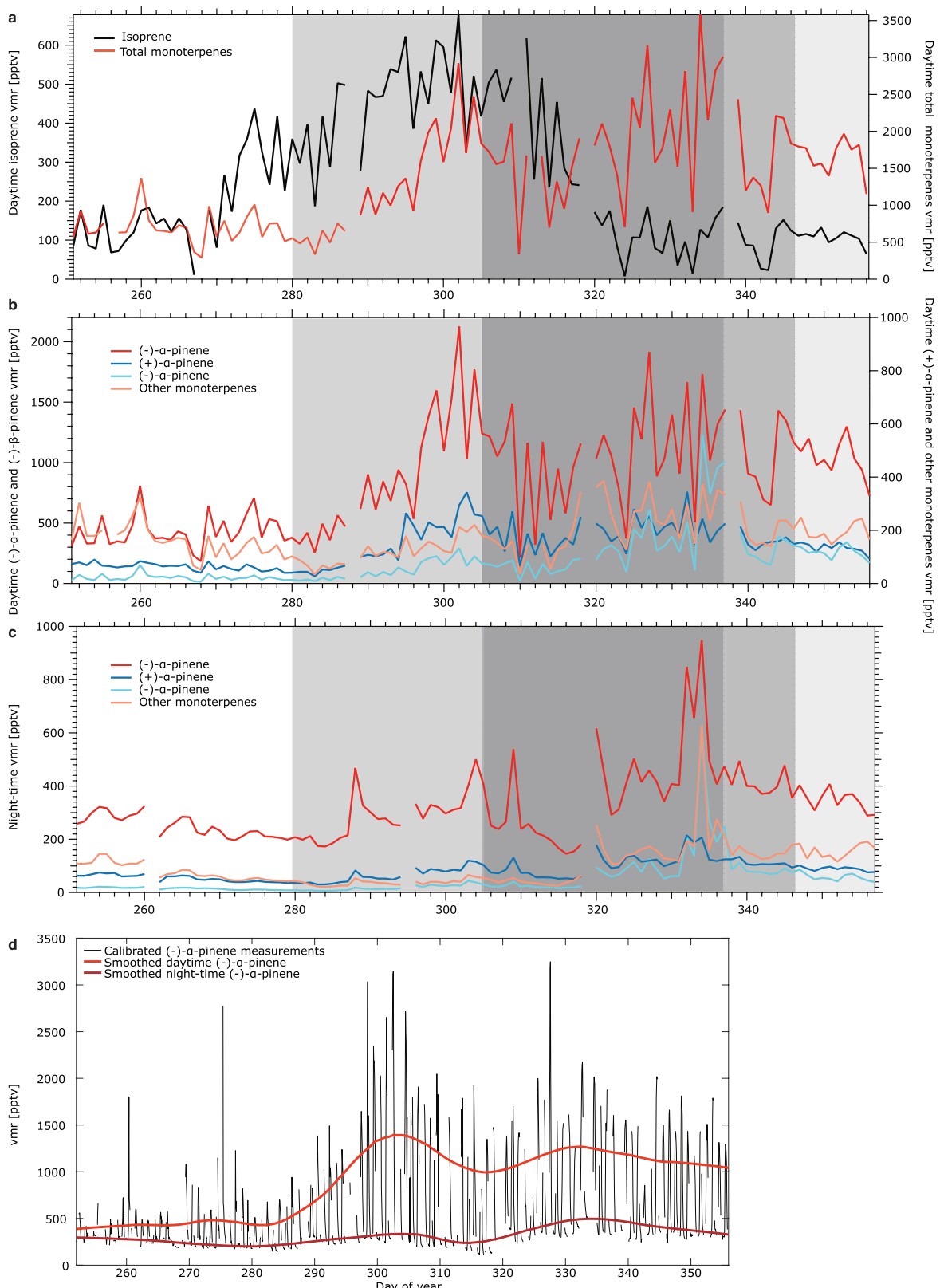

**Extended Data Fig. 6 | Daily daytime average concentrations and smoothed average plots. a**, Isoprene and total monoterpenes. **b**, (−)-α-pinene, (+)-α-pinene, (−)-β-pinene and the total of the other monoterpenes. **c**, Daily night-time average concentrations of (−)-α-pinene, (+)-α-pinene, (−)-β-pinene and the total of the other monoterpenes. **d**, Calibrated (−)-α-pinene measurements (black) with smoothed daytime (red) and night-time (brown) trend lines. The smoothed lines were created by applying a Savitzky–Golay filter in conjunction with a moving-average filter, as described in the 'Data management' section.

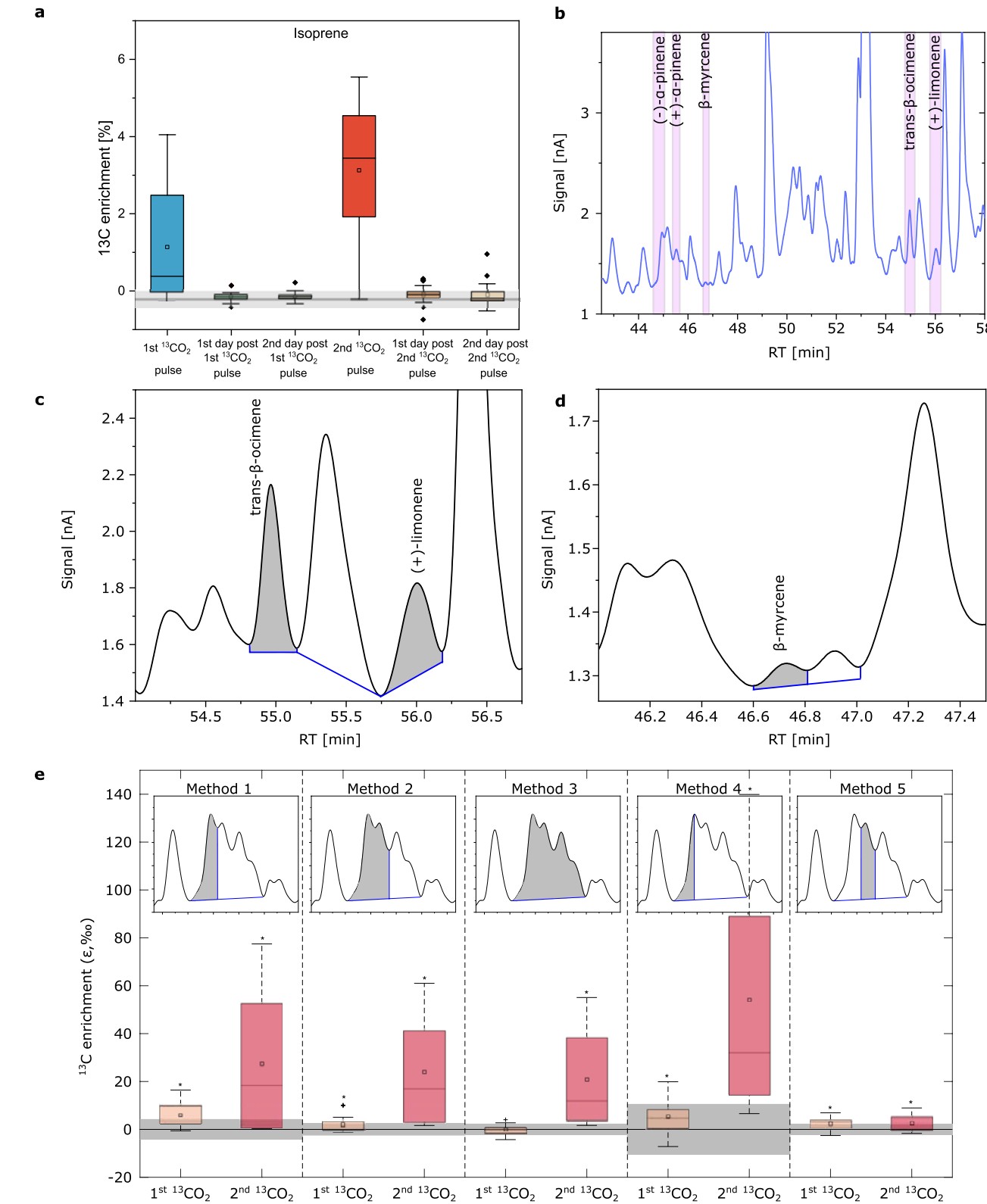

**Extended Data Fig. 7 | $^{13}$C enrichment of isoprene and (−)-α-pinene.**
**a**, $^{13}$C enrichment of isoprene measured with PTR-TOF-MS shows substantial enrichment above pre-pulse levels on the day of the labelling pulse but not on the days following the labelling pulse. **b**, A section of the chromatogram obtained from the GC-IRMS showing the peaks for the identified compounds. **c**, A section of the chromatogram obtained from the GC-IRMS showing the trans-β-ocimene peaks. The integrated regions are shaded in grey. **d**, A section of the chromatogram obtained from the GC-IRMS showing the β-myrcene peak. The integrated region is shaded in grey. **e**, Box plots representing the $^{13}$C enrichment of (−)-α-pinene. Each method represents a different way of integrating the (−)-α-pinene peak in the GC-IRMS chromatogram. The integration method is depicted in the subplot above the box plots. For **a** and **e**, the grey boxes represent the standard deviation of the values of the compounds in ambient air when there is no $^{13}CO_2$ pulse. The black line through the grey boxes represents the mean. The box plots present the median, and 25th and 75th percentiles. The small squares represent the mean and the whiskers represent the maximum and minimum acquired data points that are not considered as outliers. Significant results are indicated by the asterisk (*) above the box (that is, results are significant if $P \leq 0.05$). Statistical information for **e** is shown in Extended Data Table 3b.

**Extended Data Table 1 | Calculated isoprene and monoterpene fluxes relative to the land surface area and tree biomass carbon**

| Drought Stage | Isoprene gC ha$^{-1}$ d$^{-1}$ | Isoprene mgC kgC$^{1}$ d$^{-1}$ | (-)-α-pinene mgC ha$^{-1}$ d$^{-1}$ | (-)-α-pinene gC kgC$^{1}$ d$^{-1}$ | (+)-α-pinene mgC ha$^{-1}$ d$^{-1}$ | (+)-α-pinene μgC kgC$^{1}$ d$^{-1}$ |
|---|---|---|---|---|---|---|
| Pre-drought | 0.52 ± 0.26 | 8.2 ± 4.0 | 1.6 ± 1.0 | 25 ± 16 | 0.04 ± 0.55 | 0.59 ± 0.80 |
| Early drought | 1.44 ± 0.65 | 22.6 ± 10.3 | 6.5 ± 2.7 | 102 ± 42 | 0.40 ± 0.43 | 6.35 ± 6.71 |
| Severe drought | 0.38 ± 0.43 | 6.1 ± 6.7 | 7.6 ± 2.2 | 120 ± 34 | 0.58 ± 0.36 | 9.15 ± 5.7 |
| Deep rewet | 0.29 ± 0.06 | 4.6 ± 1.0 | 9.7 ± 1.1 | 153 ± 18 | 0.31 ± 0.02 | 4.96 ± 0.34 |
| Recovery | 0.54 ± 0.45 | 8.6 ± 7.1 | 10.4 ± 6.9 | 164 ± 108 | 0.19 ± 0.18 | 2.96 ± 2.86 |

| Drought Stage | (-)-β-pinene mgC ha$^{-1}$ d$^{-1}$ | (-)-β-pinene gC kgC$^{1}$ d$^{-1}$ | (-)-limonene mgC ha$^{-1}$ d$^{-1}$ | (-)-limonene gC kgC$^{1}$ d$^{-1}$ | (+)-limonene mgC ha-1 d$^{-1}$ | (+)-limonene μgC kgC$^{1}$ d$^{-1}$ |
|---|---|---|---|---|---|---|
| Pre-drought | 0.16 ± 0.15 | 2.5 ± 2.3 | 0.16 ± 0.20 | 0.25 ± 3.23 | 0.04 ± 0.03 | 0.66 ± 0.42 |
| Early drought | 0.55 ± 0.28 | 8.6 ± 4.5 | 0.30 ± 0.17 | 4.69 ± 2.73 | 0.01 ± 0.02 | 0.21 ± 0.35 |
| Severe drought | 3.20 ± 3.43 | 50.3 ± 54.0 | 0.37 ± 0.27 | 5.88 ± 2.14 | 0.05 ± 0.08 | 0.78 ± 1.31 |
| Deep rewet | 3.72 ± 4.48 | 58.6 ± 70.6 | 0.24 ± 0.17 | 3.73 ± 2.68 | 0.09 ± 0.01 | 1.38 ± 0.06 |
| Recovery | 1.88 ± 0.44 | 29.7 ± 6.9 | 0.02 ± 0.06 | 0.36 ± 0.94 | 0.07 ± 0.11 | 1.09 ± 1.76 |

| Drought Stage | (-)-camphene mgC ha$^{-1}$ d$^{-1}$ | (-)-camphene gC kgC$^{1}$ d$^{-1}$ | (+)-camphene mgC ha$^{-1}$ d$^{-1}$ | (+)-camphene gC kgC$^{1}$ d$^{-1}$ | γ-terpinene mgC ha-1 d$^{-1}$ | γ-terpinene μgC kgC$^{1}$ d$^{-1}$ |
|---|---|---|---|---|---|---|
| Pre-drought | -0.004 ± 0.006 | -0.06 ± 0.09 | -0.01 ± 0.01 | -0.17 ± 0.16 | 0.08 ± 0.06 | 1.3 ± 0.9 |
| Early drought | 0.009 ± 0.007 | 0.14 ± 0.11 | 0.03 ± 0.02 | 0.53 ± 0.23 | 0.18 ± 0.10 | 2.8 ± 1.6 |
| Severe drought | 0.020 ± 0.006 | 0.31 ± 0.10 | 0.11 ± 0.07 | 1.75 ± 1.07 | 0.96 ± 1.48 | 15.1 ± 23.2 |
| Deep rewet | 0.002 ± 0.002 | 0.03 ± 0.03 | 0.07 ± 0.01 | 1.06 ± 0.21 | -0.08 ± 0.28 | -1.3 ± 4.3 |
| Recovery | 0.002 ± 0.013 | 0.03 ± 0.20 | 0.06 ± 0.08 | 0.92 ± 1.30 | 0.02 ± 0.01 | 0.4 ± 0.2 |

The fluxes were calculated on the basis of the change of moles in the system, the moles exchanged with the flow through and the soil uptake rate calculated on the basis of the rate of change in moles during early night-time hours relative to the gas concentration (assuming that the gas concentration has a strong influence on the uptake rate). The uncertainty values given are 95% confidence intervals for the daily mean flux of each drought stage.

**Extended Data Table 2 | Average values of $CO_2$, vapour pressure deficit (VPD) and temperature (*T*) for the understory and the canopy over the different phases of the measurement campaign**

**a**

| Drought Stage | CO₂ [ppm] understory | CO₂ [ppm] canopy | VPD understory | VPD canopy | T [ºC] understory | T [ºC] canopy |
|---|---|---|---|---|---|---|
| Pre-drought | 472±20 | 456±19 | 0.25±0.01 | 0.83±0.13 | 22.4±0.6 | 26.8±0.6 |
| Early drought | 445±8 | 429±7 | 0.81±0.22 | 1.39±0.21 | 24.8±1.4 | 27.8±1.1 |
| Severe drought | 458±6 | 440±6 | 1.39±0.24 | 1.75±0.22 | 24.6±0.7 | 26.5±1.0 |
| Deep rewet | 451±13 | 433±13 | 1.19±0.09 | 1.45±0.10 | 24.1±0.5 | 25.7±0.4 |
| Recovery | 453±10 | 433±11 | 1.10±0.16 | 1.41±0.20 | 24.8±0.5 | 26.8±0.7 |

**b**

| Drought Stage | CO₂ [ppm] understory | CO₂ [ppm] canopy | VPD understory | VPD canopy | T [ºC] understory | T [ºC] canpopy |
|---|---|---|---|---|---|---|
| Pre-drought | 388±22 | 364±21 | 0.27±0.02 | 1.42±0.24 | 23.1±0.8 | 30.8±0.8 |
| Early drought | 402±11 | 380±11 | 0.98±0.14 | 2.22±0.16 | 26.8±0.8 | 32.4±0.9 |
| Severe drought | 434±5 | 411±6 | 1.61±0.18 | 2.30±0.34 | 26.6±1.0 | 29.8±1.6 |
| Deep rewet | 417±3 | 394±4 | 1.11±0.14 | 1.78±0.21 | 26.0±1.0 | 29.5±2.6 |
| Recovery | 405± | 380±18 | 1.18±0.12 | 1.62±0.29 | 26.8±0.7 | 29.3±1.3 |

**a**, Whole-day averages. **b**, Daytime averages. The uncertainties on the $CO_2$ values are 95% confidence intervals for the daily mean values for each drought stage. The uncertainties for the VPD and temperature are the 95% confidence intervals for the midday (11:00–15:00 local time) maximum for each drought stage.

**Extended Data Table 3 | Statistical information from *t*-test that was performed on the acquired data from the $^{13}$C-labelled atmospheric samples**

**a**

| Compound | Phase | n | DoF | t-values |
|---|---|---|---|---|
| (-)-alpha-pinene | PP | 30 | 29 | |
| | 1st pulse | 16 | 15 | 0.0005 |
| | Post 1st pulse day 1 | 6 | 5 | 0.222 |
| | Post 1st pulse day 2 | 5 | 4 | 0.1563 |
| | 2nd pulse | 11 | 10 | 0.004 |
| | Post 2nd pulse day 1 | 4 | 3 | 0.421 |
| | Post 2nd pulse day 2 | 5 | 4 | 0.112 |
| (+)-alpha-pinene | PP | 34 | 33 | |
| | 1st pulse | 16 | 15 | 0.1914 |
| | Post 1st pulse day 1 | 6 | 5 | 0.1561 |
| | Post 1st pulse day 2 | 5 | 4 | 0.3994 |
| | 2nd pulse | 9 | 8 | 0.4483 |
| | Post 2nd pulse day 1 | 4 | 3 | 9.945E-06 |
| | Post 2nd pulse day 2 | 5 | 4 | 0.4574 |
| (-)-beta-pinene | PP | 23 | 22 | |
| | 1st pulse | 15 | 14 | 0.1084 |
| | Post 1st pulse day 1 | 6 | 5 | 0.4687 |
| | Post 1st pulse day 2 | 5 | 4 | 0.2137 |
| | 2nd pulse | 11 | 10 | 0.0027 |
| | Post 2nd pulse day 1 | 4 | 3 | 5.966E-07 |
| | Post 2nd pulse day 2 | 4 | 3 | 0.3705 |
| trans-beta-ocimene | PP | 33 | 32 | |
| | 1st pulse | 16 | 15 | 0.094 |
| | Post 1st pulse day 1 | 6 | 5 | 0.3513 |
| | Post 1st pulse day 2 | 5 | 4 | 0.0370 |
| | 2nd pulse | 10 | 9 | 0.0055 |
| | Post 2nd pulse day 1 | 4 | 3 | 0.0005 |
| | Post 2nd pulse day 2 | 5 | 4 | 0.0365 |
| (+)-limonene | PP | 26 | 25 | |
| | 1st pulse | 15 | 14 | 0.4563 |
| | Post 1st pulse day 1 | 5 | 4 | 0.1737 |
| | Post 1st pulse day 2 | 4 | 3 | 0.4713 |
| | 2nd pulse | 11 | 10 | 0.0494 |
| | Post 2nd pulse day 1 | 4 | 3 | 0.0441 |
| | Post 2nd pulse day 2 | 5 | 4 | 0.1091 |

**b**

| Method | Phase | n | DoF | t-values |
|---|---|---|---|---|
| 1 | PP | 29 | 28 | |
| | 1st pulse | 16 | 15 | 0.0003 |
| | 2nd pulse | 11 | 10 | 0.0038 |
| 2 | PP | 30 | 29 | |
| | 1st pulse | 16 | 15 | 0.0221 |
| | 2nd pulse | 10 | 9 | 0.0028 |
| 3 | PP | 29 | 28 | |
| | 1st pulse | 16 | 15 | 0.4502 |
| | 2nd pulse | 11 | 10 | 0.0026 |
| 4 | PP | 36 | 35 | |
| | 1st pulse | 14 | 13 | 0.02 |
| | 2nd pulse | 11 | 10 | 0.0016 |
| 5 | PP | 15 | 14 | |
| | 1st pulse | 14 | 13 | 0.0036 |
| | 2nd pulse | 10 | 9 | 0.0222 |

DoF is the degrees of freedom and *n* is the number of samples. PP refers to the samples taken before the $^{13}CO_2$ labelling (pre-pulse), 1st and 2nd pulse refer to the first and second $^{13}CO_2$ pulses, respectively, and Post 1st pulse day 1 and Post 1st pulse day 2 refer to the days immediately following the day of the $^{13}CO_2$ pulses. **a**, All monoterpenes. **b**, The different methods of integrating (−)-α-pinene (Extended Data Fig. 7e). DoF is the degrees of freedom and *n* is the number of samples. PP refers to the samples taken before the $^{13}CO_2$ labelling (pre-pulse) and 1st and 2nd phases refer to the first and second $^{13}CO_2$ pulses, respectively.