## [Peer Review File · Nature]

Manuscript Title: Chiral monoterpenes reveal forest emission mechanisms and drought responses.

Reviewer Comments & Author Rebuttals

Reviewer Reports on the Initial Version:

Referees' comments:

Referee #1 (Remarks to the Author):

A Summary of the key results:

The paper describes in detail the dynamical emission patterns of chiral monoterpenes at a drought experiment in a tropical mesocosm. The reasoning and justification of the study are well explained and clear - there is need to understand better both the plant BVOC biosynthesis and emission pathways and their diversity, and the feedbacks to the atmosphere and climate especially in a stressed ecosystem. Stress factors like drought are increasing in abundance even in the tropics, and may affect strongly to the ecosystem scale emissions and thus the ecosystem-atmosphere interactions. It is evident that the BVOC emission responses to drought are still rather poorly understood, and that more research is needed to clarify the often contradictory results of drought on BVOC (especially monoterpenes) emissions.

This study sheds light to the mechanisms of feedbacks between plants and drought stress. The results indicate a potentially important molecular response of the α -pinene emissions, which may be due to the specific enzymatic biosynthesis pathway or the storage dynamics of the different pinene enantiomers. The emission patterns responded differently to the progressive drought, both in diurnal and longer time scales. Isotope labelling indicated that in normal conditions, (-)- α -pinene was emitted from a fast and dynamic pool while the (+)- α -pinene was not labelled and thus was concluded to originate from a more stable storage. In severe stress conditions (as photosynthesis was limiting the carbon source for biosynthesis), also the de novo produced (-)- α -pinene source was shifted to the storage pool. This shows that there are previously unknown or poorly understood feedbacks between stressed ecosystems and atmosphere, which may be important for our understanding of climate change.

B. Originality and significance:

The experiment and data are definitely original and of high value for a better process understanding of the biosynthesis and emission of BVOCs from tropical ecosystems. This approach is crucially needed in the biosphere-atmosphere exchange perspective, and the presented data creates a clear link between the detailed mechanistic plant studies in a near-real life conditions and the atmospheric consequences. 'Near-real' because mesocosm is still an artificial system and limited in space and constrains many biological and physical factors which need to be taken into account as much as possible; especially here the lack of any atmospheric reactivity measurements is disturbing.

As such, the results are likely not of much immediate interest to a general scientific community – to my view this ms would definitely be suitable for publication in the best journals of the discipline but not for Nature, unfortunately. Also after seeing the accompanying, related manuscript, I feel this paper is too slim to warrant a stand-alone publication without the details e.g. on functional

group -specific drought responses provided in the other paper.

C. Data & methodology: validity of approach, quality of data, quality of presentation:

As such, the experiment is robust and both the drought experiment and consequent isotope labelling and measurements have been executed in a highly professional manner. The results are clear and analysis has been done carefully and with appropriate statistical tools. There are no such flaws that would prohibit the publication; however several remarks can be made on the data provided and the conclusions drawn on the basis of the data.

- The authors claim the enclosure conditions do not allow photochemistry to take place, however they do not present data to back this up. Were O₃, NO₃ or OH concentrations measured during the experiment?

- In nature, drought is almost always accompanied with hot temperatures. This was not discussed nor any analysis of the temperature-dependency of emissions was done.

- No labelling patterns of isoprene are presented. This would support the conclusion of the response of de-novo synthesised (-)- α -pinene synthesis. As isoprene is not stored in tissues, its labelled emissions should clearly show the proportion of carbon from recent photosynthates and it can be compared with the labelled monoterpenes. Such an analysis was done e.g. in Ghirardo et al (2010) who presented a hybrid algorithm for analysing the ecosystem scale emissions. At least an attempt to do such analysis from the isotopic results is needed.

- More clarifications of the ecosystem composition and the plant (both trees and understorey) species would be needed; did the authors consider that the rainforest species may differ largely in their drought resilience and the emission dynamics (some emit only isoprene, some only monoterpenes, and some a mixture of variable composition)? I note that there is more explanation in the accompanying paper on that matter.

D. Appropriate use of statistics and treatment of uncertainties:

The presented statistical analyses are rather straightforward and simple, so no criticism on that. However, there is no attempt to e.g., create a statistical model for better describing the dynamic emissions, or to compare with the currently existing models for emissions. With such modelling work added, the paper would definitely be more valuable for specialists and for wider audiences.

E. Conclusions: robustness, validity, reliability:

The authors correctly conclude that more detailed emission (enantiomeric resolved) measurements are required to accurately predict the ecosystem responses to droughts. Such measurements can indeed provide deeper insights on the response mechanisms, which are highly valuable and will increase our understanding on the feedbacks between ecosystems and atmosphere. However, it is a big question if and where such measurements can be realistically implemented, and what would be their value to larger scale predictions of the impacts of extreme drought events.

F. Suggested improvements: experiments, data for possible revision:

The data on soil uptake would be needed for a proper understanding of the bidirectional exchange of monoterpenes. Were there any isotopic measurements done from the soil chambers?

How did the enantiomeric fractionation of VOC uptake in the soil change and did that affect the results (especially as drought is a major disturbance for the soil microorganisms at least partially responsible for the uptake)?

Data on changes in VPD instead or in addition of the Atmospheric Water Potential would have been needed.

No analysis on the temperature impacts is included, that would be important for understanding the environmental drivers better.

H. Clarity and context: lucidity of abstract/summary, appropriateness of abstract, introduction and conclusions:

The ms is written basically in good language, some typos are remaining which can easily be corrected. Abstract is clear and summarizes the main results; however the indication of the results for atmospheric chemistry without any photochemistry data is not appropriate.

Referee #2 (Remarks to the Author):

The manuscript by Byron et al. describes observational data on emissions of chiral monoterpenes from the Biosphere II. While the emission of chiral monoterpenes has long been recognized, the reported findings are novel and highlight the need to improve BVOC emission models. Abstract and summary are well written and summarize the findings adequately. The impact of drought plays a major role in modulating BVOC emissions, and as such the presented results are an important piece of information still missing when modeling monoterpene emissions. Data and methodology are adequately described, but there are some discrepancies with the accompanying paper that should be explained. Also, more detailed information on species distribution would be beneficial in order to better understand the ecological significance of the findings for a rainforest growing in the real world. It is mentioned that 95 tropical species are grown in B2. What is the relative distribution of biomass? What is known about monoterpene emissions from these species (ie. how many of the species actually emit MT)? Can the results be distilled to a couple of high MT emitting species or are they more representative for the entire ecosystem? This also relates to the connected interpretation of Fig3 and associated modelling exercises. Why can it be assumed that chiral monoterpenes are exclusively emitted internally mixed (i.e. de novo and stored (only) from the same plant)? Couldn't it just be that different species emit different chiral monoterpenes, and the actual effect seen is that some species see largely different responses to drought stress than other species? Deciphering the relative distribution of MT emitting species and their individual response to the drought would be important to support the underlying conclusion of this study. During severe drought stress one would also expect other VOCs emitted (e.g. green leaf volatiles, or compounds from the shikimate pathway)? Is there any additional evidence from other 'marker species', and how do their emission patterns during the drought correspond to the changing chiral pattern of MT emissions?

What was the exchange rate of air during the experiment? Some details are still unclear concerning sources and sinks of BVOC. In the accompanying paper the authors state that "Due to the UV-screening glass of Biosphere 2, atmospheric oxidation effects are negligible and atmospheric VOC concentrations reflect the interplay between vegetation emissions and net exchange rates with soil". This seems too simplified to me – wouldn't the exchange rate through the air handling system also be a major sink due to mixing with 'fresh outside air', or how was this accounted for (e.g. using SF6 data) and/or factored in the data analysis? It would be good to know how fast the air was exchanged, as it could serve to determine actual net exchange rates, from which net emissions per unit biomass could be calculated. This could at least help comparing B2 emissions with those obtained in real-world ecosystems in the tropics. How similar or different are these? Putting results more in context of the real world would help strengthen the ecological significance of the conducted experiments. Isoprene concentrations for example reached levels up to 500 ppbv. Those of total monoterpenes only 2 ppbv. This is only about 0.4%. Typical MT emissions over tropical forests are quite a bit higher relative to isoprene. Is there an explanation based on MT emitting species in the B2? Also, in the accompanying Science paper quite a bit lower isoprene concentrations were reported for the same (?) period (e.g. ~500 ppbv on day 300 in this

paper vs. ~150 ppbv on day 300 in the paper submitted to Science). Or was this a different year – if so what were the differences?

Minor comments:

Figure 2 shows some sort of model smoothed BVOC concentrations, in my opinion it would be beneficial to add actually measured concentrations in this figure as well (e.g. aggregated daily means or similar). What is the shaded modelled line (measured uncertainty or real daily variability in the B2)?

Referee #3 (Remarks to the Author):

Ambient air concentrations of monoterpenes were monitored throughout a drought experiment in a tropical forest ecosystem enclosure. Changing relative contribution of diverse monoterpenes to whole system total monoterpene emissions during the experiment were observed as well as different labeling intensity of enantiomers of α -pinene and different shifts in diel cycles of diverse enantiomers in response to the drought treatment. The results evidence the role of differences in residence times of diverse emission sources.

I suggest to change the title of the paper, as the results evidence the relevance of emissions from sources of different residence times rather than reveal mechanisms.

Several bits of additional information needed to fully understand the experiment are communicated with the companion paper only, as e.g. location of Biosphere 2, height of the enclosure, non-use of additional lighting, time course of $\delta^{13}\text{C}$ of the atmosphere, operations applied to impose early and severe drought. They should be added to the text and extended data. The statement in the Methods section line 52 likely is true. However, the basis of this assumption should be explicated with one phrase and if possible a reference added.

Sampling was performed at the height where the canopy is most dense. Can information on within system gradients (CO_2 , water vapor) be supplied?

Can supporting information on the absence of potential reactants for monoterpenes in the enclosure air be given?

Show the time course of air $\delta^{13}\text{C}$ measurements during and after the labeling pulses with a figure in the extended data (Methods section lines 147-148). The information available with the companion paper indicates that ^{13}C label returned to the air of the enclosure in significant amounts for several days due to respiration of diverse ecosystem components. Thus, though the enclosure was ventilated at the end of the pulses, photosynthesis used weakly labelled source CO_2 for several days.

There are inconsistencies between the dates for transition from early drought to severe drought and end of severe drought between figure 2 and figure 2 of the companion paper. In addition at line 103 peaks in monoterpene mixing ratio after 33 days and after 66 days are mentioned, which according to figure 2 rather occur at 23 days and 56 days, respectively.

The description of the sensitivity analyses in lines 197-198 for the quantification of (single) peaks in the GC-IRMS chromatograms shown with Fig. 6 in the extension data does not reflect what has been shown with the figure.

Suggestions for refocusing the paper:

The correlation of α -pinene enantiomer concentrations with other MT (line 26) is not communicated with the paper.

Though the physical properties of α -pinene enantiomers do not differ much (Lines 34-35), a recent study showed differences in the physical properties of derived dimers: Bellcross A., Bé A.G., Geiger F.M. & Thomson R.J. (2021) Molecular Chirality and Cloud Activation Potentials of Dimeric α -Pinene Oxidation Products. Journal of the American Chemical Society.

I suggest to add the description of the dynamics of the ratio (parallel rise and earlier peak in isoprene) through the early drought period to lines 129-132.

The text on lines 177-180 provides a misleading description. The conclusion can be drawn for the α -pinene only as parameterization of the light and temperature response functions does differ between different monoterpenes.

More information on the post-pulse dynamics of e13C, only addressed with a summary statement (lines 149-150), should be given, as the storage issue is a central topic of the paper. In addition add information on the time frame within which post-pulse samples were taken to line 157 in the methods section.

I suggest to more deeply explore the night time and long-term emission effects of labeling recognized as largely unknown terrain by Niinemets & Reichstein.

The conclusion that the results „show“ (L229) that changes in fractional contribution of monoterpenes can be used to gauge drought in tropical ecosystems is a bit premature. It rather is a suggested hypothesis that requires further testing.

The conclusion on identification of sources and changes in carbon cycling (L234-236) is ambiguous. Rather the measurements indicated the existence of source pools with different half-life times and changes in their relative contribution to emissions in response to drought. However, the identity of the pools within the complex ecosystem comprising many different species, differentially exposed to light and VPD during the day, potential translocation and absorption by different plant organs can not be deduced from the presented results.

Finally the conclusion about the need to include enantiomerically resolved monoterpenes in large scale models (L240-242) is far to blunt. The relevance of this step has not been studied with the current paper. The general request for "accurately assessing" has no scientific sense. No model is a one-to-one digital representation of reality. The accuracy that can be achieved with the simplifying assumptions of model structures needs to be gauged against specific objectives, and for specific target values.

Emission models require a fair number of experimental studies that provide results on emission factors in order to parameterize the emission factors (EF). The presentation of the updating of EF between two model versions in Messina et al. 2016 illustrates well the importance of this fact.

Thus, prior to distinguishing between chiral forms the number of experimental results needs to be increased. This issue is further underlined by the differences between emissions simulated with the ORCHIDEE and MEGAN models shown in the same paper, indicating the relevance of other factors (e.g. number and distributions of plant functional types, scarcity of experimental information for parameterizing the light response) impacting the flux calculations.

Furthermore, as shown by Niinemets and Reichstein 2002, further refinement of models requires to include storage pools as emission rates probably are rarely in steady state. However, they also recognized that including further detail into the models applied at large scales may not improve their performance. Temporal variation in emission of diverse compounds plays an important role. How important is the single factor of chiral forms?

I suggest to focus more on discussion of the ecosystem components potentially involved in causing the observed changes in monoterpene emissions and progress made relative to the papers on monoterpene emission including enantiomers cited in the introduction instead of repeatedly making strong statements about consequences for larger scale modelling.

Suggestions for editing:

L17 why „but“

L20/21 instead of „without photochemistry“ better „in absence of UV light“

L23 emitted (-)- α -pinene mainly de novo → emitted mainly de novo synthesized (-)- α -pinene

L24 (-)- α -pinene emissions → the source of (-)- α -pinene emissions

L25/26 The α -pinene enantiomers each → Mixing ratios of both α -pinene enantiomers

L60 compound → compounds

L60-62 Phrase incomprehensible. What do you mean by intractable?

L66 delete „measuring“

L97 Is this an averaged over the whole periods?

L131 This means that → Thus

L157 Rather: „.. (-)- α -pinene trans- β -ocimen and β -myrcene synthesized with photosynthetic assimilates produced during the labeling pulse were directly emitted ...”

L160-162 awkward wording, rephrase

L197 from the → from storage in the; emissions fluxes most likely have to transit the aqueous phase

L199 quantities → fractions

L224 when → with

Methods

L47 delete „be”

L176 the H₂O → of H₂O

Author Rebuttals to Initial Comments:

General (common themes from the reviewers)

- 1) A common theme for all reviewers was the request for improved justification for the statement that oxidation chemistry was absent in the Biosphere 2. We now expand on why this is so. The incoming air passes through large air handling units, thereby largely removing ozone due to its sensitivity to surfaces. Additionally, the glass from which the Biosphere 2 is constructed does not transmit light at wavelengths that generate OH. Thus significant oxidation by ozone, OH or NO₃ (which requires ozone to form) cannot occur. We substantiate the point by reference to measured ratios of isoprene and isoprene oxidation products, which are 100 times richer in isoprene than measured in the outdoor Amazon rainforest. Furthermore, since the OH oxidation of isoprene is much faster than for monoterpenes, the absence of atmospheric chemistry should result in a lower monoterpene to isoprene ratio compared to the real-world, which is exactly what we observe. In light of the reviewer's comments, we have also made ozone measurements, which found a very low amount, on average 1.1 ± 0.7 ppb, at a central location within the Biosphere 2 tropical rainforest compared with an average of 49.2 ± 1.2 ppb outside the Biosphere 2. We now add these comparisons to Amazon rainforest measurements to make this point clearer to the reader. For completeness, we also expanded upon the description of the Biosphere 2 including dimensions, flushing rates, and confirming the absence of artificial lighting (which could have provided a photolysis source).
- 2) A second common theme was the request for more detail concerning the vegetation, and differing responses among different vegetation types. To address this, we now include in the methods section a much more comprehensive description of the vegetation present in the Biosphere 2. Furthermore, we have added enantiomerically resolved monoterpene data taken from two predominant species in the Biosphere 2, a canopy species (*C. fairchildiana*) and an understory species (*Piper* sp.) both of which emit isoprene and monoterpenes. These cuvette-based measurements were taken for two of the most abundant plant species periodically throughout the entire measurement campaign. The data shows that different individual plants respond differently and that what we present is the overall ecosystem response for this mesocosm. We also now note that some species present (*Pachira aquatica*) only emit isoprene. In addition, we make more reference to the companion ecological

description paper by Werner et al. 2021, which comprehensively describes the facility and is now published in *Science*.

- 3) A final theme addressed by two reviewers is temperature effects and the relation to current emission models. In the extended data we now include, emission profiles predicted by the current state-of-the-art emission model MEGAN at various stages of the drought (different temperatures) and those of the measured enantiomerically separated monoterpene data. During the drought, they are markedly different, both in absolute amount and in the diel temporal profile. This helps highlight the importance of this work, namely that current temperature and light-based approaches to emission modelling will not be able to reproduce the observed responses as both enantiomers are modelled together erroneously assuming they behave identically. A deeper, more plant process-based approach is justified on the basis of these and other recently published results (Messina et al 2016). With regard to atmospheric impacts, the shift in the time of the emission is particularly important, as this will affect processes related to the formation of cloud condensation nuclei. This is because towards afternoon there is a shift in the partitioning between evaporation and sensible heat flux that favors the latter. More sensible heat flux means more turbulence in the afternoon, which would facilitate vertical transport of the later emitted species to more oxidative, cooler regions.

Referees' comments:

Referee #1 (Remarks to the Author):

A Summary of the key results:

The paper describes in detail the dynamical emission patterns of chiral monoterpenes at a drought experiment in a tropical mesocosm. The reasoning and justification of the study are well explained and clear - there is need to understand better both the plant BVOC biosynthesis and emission pathways and their diversity, and the feedbacks to the atmosphere and climate especially in a stressed ecosystem. Stress factors like drought are increasing in abundance even in the tropics, and may affect strongly to the ecosystem scale emissions and thus the ecosystem-atmosphere interactions. It is evident that the BVOC emission responses to drought are still rather poorly understood, and that more research is needed to clarify the often contradictory results of drought on BVOC (especially monoterpenes) emissions.

This study sheds light to the mechanisms of feedbacks between plants and drought stress. The results indicate a potentially important molecular response of the α -pinene emissions, which may be due to the specific enzymatic biosynthesis pathway or the storage dynamics of the different pinene enantiomers. The emission patterns responded differently to the progressive drought, both in diurnal and longer time scales. Isotope labelling indicated that in normal conditions, (-)- α -pinene was emitted from a fast and dynamic pool while the (+)- α -pinene was not labelled and thus was concluded to originate from a more stable storage. In severe stress conditions (as photosynthesis was limiting the carbon source for biosynthesis), also the de novo produced (-)- α -pinene source was shifted to the storage pool. This shows that there are previously unknown or poorly understood

feedbacks between stressed ecosystems and atmosphere, which may be important for our understanding of climate change.

We thank the reviewer for eloquently summarizing our work and highlighting its importance in the global context.

B. Originality and significance:

The experiment and data are definitely original and of high value for a better process understanding of the biosynthesis and emission of BVOCs from tropical ecosystems. This approach is crucially needed in the biosphere-atmosphere exchange perspective, and the presented data creates a clear link between the detailed mechanistic plant studies in a near-real life conditions and the atmospheric consequences. 'Near-real' because mesocosm is still an artificial system and limited in space and constrains many biological and physical factors which need to be taken into account as much as possible; especially here the lack of any atmospheric reactivity measurements is disturbing.

As such, the results are likely not of much immediate interest to a general scientific community – to my view this ms would definitely be suitable for publication in the best journals of the discipline but not for Nature, unfortunately. Also after seeing the accompanying, related manuscript, I feel this paper is too slim to warrant a stand-alone publication without the details e.g. on functional group-specific drought responses provided in the other paper.

This is a surprising view, particularly following the reviewer's previous remarks in the summary. With this work, we clearly show that atmospheric monoterpenes, which are important for particle formation and associated global radiative effects are currently measured and modelled incorrectly, especially with regard to ecosystem drought responses. More specifically, we show that by measuring both enantiomers together (as is the current accepted practice) there is no chance to correctly simulate drought responses of tropical ecosystems, since the enantiomers do vary independently (in both timing, strength, and production mechanism). These findings could have only been discovered through use of this unique experimental facility, with its enclosed tropical rainforest mesocosm, the meteorologically controlled environment, and the isotopic labelling capability. This work therefore represents a paradigm change in the way such compounds should be measured, in how emission inventories should be constructed, and how monoterpene emissions are modelled. It is shown that enantiomers need to be considered separately by atmospheric scientists despite their equivalent rates of atmospheric oxidation, as they are controlled by different emission processes. Furthermore, we show that these complex enzymatically controlled emission processes can be monitored by measuring enantiomers in the air. We feel that this is a breakthrough in our understanding of previously observed chiral changes and in how to think about these species as atmospheric scientists, otherwise we would not have submitted the work to this journal.

We are confused why the lack of any atmospheric reactivity measurements is "disturbing". We have in the past made OH reactivity measurements in real-world Amazon tropical forest environments as a budgeting exercise, initially highlighting missing OH reactivity (Nölscher et al. 2016 Nature Communications) and then later closing the budget with improved measurement technology for oxygenated species (Pfannerstill et al. ACP 2021). We have also previously noted changes in the OH reactivity profile in response to El Niño conditions (Pfannerstill et al. 2018). However, OH reactivity

measurements are not useful in the context of this experiment (which is why we did not make this measurement here), nor are they relevant to the main results presented here. The concept of OH reactivity is useful in assessing the dynamic response of the OH loss frequency, either together with comprehensive measurements to assess closure or together with OH sources to determine OH radical abundances at steady state. In the Biosphere 2, as stated in the script, there is no atmospheric oxidation chemistry present (see general comments above point 1). The OH reactivity of the air in the Biosphere 2 will be completely dominated by the isoprene concentrations which are much higher than real-world environments precisely because there is no chemical sink present. The focus here is on the enantiomeric monoterpenes which contribute minimally to the OH reactivity in the Biosphere 2. In the absence of complicating variable factors like meteorology, atmospheric chemistry, and advected emissions, the results from this unique facility reveal how plants change their emission profiles in response to drought, namely that enantiomers vary independently of one another.

C. Data & methodology: validity of approach, quality of data, quality of presentation:

As such, the experiment is robust and both the drought experiment and consequent isotope labelling and measurements have been executed in a highly professional manner. The results are clear and analysis has been done carefully and with appropriate statistical tools. There are no such flaws that would prohibit the publication; however several remarks can be made on the data provided and the conclusions drawn on the basis of the data.

- The authors claim the enclosure conditions do not allow photochemistry to take place, however they do not present data to back this up. Were O_3 , NO_3 or OH concentrations measured during the experiment?

No significant atmospheric chemistry was possible in the biosphere. Large air handling units largely remove ozone contained with fresh incoming air due to surface losses, and glass does not transmit light at wavelengths that generate OH (Finn et al. 1999). Thus, significant oxidation by ozone, OH or NO_3 cannot occur. This is consistent with the ratio of isoprene to its oxidation products which is 100 times richer in isoprene than typically measured in the Amazon rainforest. It is also supported by the ratio of isoprene to monoterpenes, which is also enriched in isoprene relative to Amazon rainforest measurements as isoprene reacts faster with OH than the monoterpenes.

In order to emphasize and substantiate this important point we now add the following text to the manuscript,

“No significant OH oxidation chemistry can occur in the B2-TRF since glass does not transmit light at wavelengths that generate OH, and ozone within fresh incoming air is lost to the surfaces of large air handling units. The absence of any significant photochemistry is reflected in the ratio of isoprene to its oxidation products (which is 100 times richer in isoprene than in typical Amazon rainforest measurements) and in the isoprene to monoterpene ratio (which favors isoprene in the Biosphere 2 by a factor of ~ 3 due to its faster OH reaction coefficient). Ozone was also measured post-campaign within the B2-TRF and found to be at very low concentrations, on average 1.1 ± 0.7 ppb whilst outside the Biosphere 2, the air was found to contain an average ozone concentration of 49.2 ± 1.2 ppb.”

- In nature, drought is almost always accompanied with hot temperatures. This was not discussed nor any analysis of the temperature-dependency of emissions was done.

The average temperature throughout the experiment is given in Figure 4 and its ecological impact discussed in the companion paper Werner et al. Science 2021. In this manuscript, it is already explained that during pre-drought some monoterpenes exhibited a stronger dependency on temperature than light and for other monoterpenes the opposite was true. It is also stated that as the drought progressed, the monoterpenes that were previously more dependent on light than temperature became more dependent on temperature since their concentrations peaked at the same time as temperature.

Nevertheless, we take the reviewer's point that temperature is a key driver and deserves more attention. Accordingly, we now include in the extended data plots a figure showing emission fluxes predicted by the MEGAN emission model for the light and temperature conditions experienced at several stages in the drought and compare them to the measured, enantiomerically separated data. This shows that during drought the emissions substantially exceed those predicted and that the modelled data do not account for the shift observed in the diel emission profile. The following text was added,

“Ext. Data Fig. 8 shows the measured enantiomers of α -pinene with the temperature and light-based predictions of the emission model MEGAN. During drought, the measured emissions substantially exceed those predicted and the modelled data do not account for the shift observed in the diel emission profile.”

- No labelling patterns of isoprene are presented. This would support the conclusion of the response of de-novo synthesised (-)- α -pinene synthesis. As isoprene is not stored in tissues, its labelled emissions should clearly show the proportion of carbon from recent photosynthates and it can be compared with the labelled monoterpenes. Such an analysis was done e.g. in Ghirardo et al (2010) who presented a hybrid algorithm for analysing the ecosystem scale emissions. At least an attempt to do such analysis from the isotopic results is needed.

Unfortunately, it was not possible to gain labelling information of isoprene from the GC-IRMS because the sorbent tubes that were used for this sampling were not suitable for trapping isoprene. However, we have isoprene labelling data measured by a PTR-ToF-MS, which is already contained within the companion paper Werner et al., 2021. A new figure shown below was made and added to the extended data to show the enrichment of ^{13}C in isoprene. This figure shows that isoprene emissions had become enriched in ^{13}C on the day of labelling but no enrichment was seen on subsequent days, similar to what was seen for (-)- α -pinene. Since it is well known that isoprene is only a product of *de novo* synthesis, we can conclude that there must be a *de novo* synthesis component to (-)- α -pinene emissions. The following text was also added to the methods section to describe the ^{13}C isoprene analysis,

“Details of isoprene analysis are provided by Werner et al. (2021). Briefly, atmospheric isoprene concentrations were determined at a height of 13 m in the rainforest. For this purpose, ambient air was sucked through ¼” heated PFA tubing to a proton-transfer-reaction time-of-flight mass spectrometer (4000ultra PTR-TOF-MS, Ionicon Analytik, Innsbruck, Austria). Measurements were

taken for 5 minutes every two hours; in these intervals, 2 min averages (minutes 2.5 to 4.5) were used after quality control. Explicit calibrations with isoprene calibration gas were performed regularly using a dilution curve which was obtained with a liquid calibration unit (LCU, Ionicon Analytik, Innsbruck, Austria). Data obtained were processed with the software package PTRwid. Labelling of isoprene with ^{13}C from $^{13}\text{CO}_2$ was calculated as the ratio of the abundance of the single ^{13}C -labelled isoprene isotope (m/z 69 analysed by PPTR-TOFMS as m/z 69) vs. total isoprene, i.e. the non-labelled isoprene isotope (m/z 68 measured as m/z 69) plus the single labelled isotope. Due to the natural abundance of ^{13}C (1.1%), the background level of the single labelled isoprene was 5.5% considering the five C atoms of the isoprene molecule. This background level was subtracted from the data shown (Ext. Data Fig. 5)."

- More clarifications of the ecosystem composition and the plant (both trees and understorey) species would be needed; did the authors consider that the rainforest species may differ largely in their drought resilience and the emission dynamics (some emit only isoprene, some only monoterpenes, and some a mixture of variable composition)? I note that there is more explanation in the accompanying paper on that matter.

We agree with the reviewer that the survival strategy/resilience of the trees and understory may differ and there could be large variations among the different plant species and that is why we present the combined ecosystem response here. As the reviewer correctly points out, there is more information in the accompanying (now accepted) paper on this matter. Nevertheless to embellish this point we now include in the extended data the time course of monoterpene emissions that were periodically measured for *C. fairchildiana* (canopy plant) and *Piper* sp. (understory plant). This shows clearly that different species respond differently to drought and that the results presented here represent the overall ecosystem response.

Accordingly, we add the following text to lines 133-136,

“The periodic measurements of monoterpene emissions from 4 *C. fairchildiana* trees and 4 *Piper* sp. plants (from cuvettes) are provided in **Ext. Data Fig. 4**. Since the individual plant responses are largely different to what was measured in the atmosphere, the atmospheric measurements reflected the net response of the well-mixed atmosphere of the ecosystem.”

D. Appropriate use of statistics and treatment of uncertainties:

The presented statistical analyses are rather straightforward and simple, so no criticism on that. However, there is no attempt to e.g., create a statistical model for better describing the dynamic emissions, or to compare with the currently existing models for emissions. With such modelling work added, the paper would definitely be more valuable for specialists and for wider audiences.

The reviewer correctly points to the logical next step, which is to establish new emission inventories based more fundamentally on enzyme groups, storage pools, rather than assumed generalized response curves. This represents a major conceptual overhaul of current methods. As pointed out by reviewer 3 this requires considerably more data, both from laboratories and field experiments from different forest types, as well as a more sophisticated conceptual framework. It is therefore beyond the scope of what can be achieved from this dataset in isolation. Our results (and those of others,

e.g., Messina et al 2016) do show, however, that the current framework for emission inventories is inadequate in the case of drought.

To better show the discrepancy between the current emission models and measured atmospheric concentrations, we have now included in the extended data plots of the predicted emission rates of monoterpenes from the main emission model (MEGAN) compared with the measured GC data. During drought the emissions substantially exceed those predicted and the modelled data do not account for the shift observed in the diel emission profile. The following text was added,

“Ext. Data Fig. 8 shows the measured enantiomers of alpha pinene with the temperature and light-based predictions of the emission model MEGAN. During drought, the measured emissions substantially exceed those predicted and the modelled data do not account for the shift observed in the diel emission profile.”

In addition, we now also address the point that a new conceptual framework for future emission models is needed, in the manuscript via the following inserted text,

“The enantiomeric results presented here concur with previous studies that have shown the current approach of relating VOC emissions simply to light and temperature is inadequate for simulating changes associated with drought. Extra levels of complexity, ideally based on real physical processes such as storage pools and enzyme models are required.”

E. Conclusions: robustness, validity, reliability:

The authors correctly conclude that more detailed emission (enantiomeric resolved) measurements are required to accurately predict the ecosystem responses to droughts. Such measurements can indeed provide deeper insights on the response mechanisms, which are highly valuable and will increase our understanding on the feedbacks between ecosystems and atmosphere. However, it is a big question if and where such measurements can be realistically implemented, and what would be their value to larger scale predictions of the impacts of extreme drought events.

These enantiomerically explicit measurements are currently being made by our research group at the measurement site ATTO in the Amazon rainforest (e.g. Zannoni et al. Nature Comms. Earth & Env. 2020). Realistic implementation of emission processes controlled by enzymes and storage pools is indeed a massive task, but one that now appears necessary based on this and other drought focused work.

Since the ambient temperature can be controlled inside the Biosphere 2, this measurement site allows for light and temperature to be decoupled from each other, which is not possible at a real-world measurement site. This means that the processes that govern the emission of monoterpenes can be more clearly revealed and emission models improved accordingly. For example, during the pre-drought (-)-alpha-pinene peaks with light whereas (+)-alpha-pinene peaks with temperature. Firstly, similar real-world measurements would not reveal this since peak light and temperature are usually no more than an hour apart. Secondly, if the alpha-pinene enantiomers were not measured separately, it would not be possible to reveal that the enantiomers actually have different dependencies on light and temperature. Thus, measurements such as these, along with many other future laboratory and field measurements can help formulate the next generation emission models with a more physiologically founded basis. In particular, the Biosphere 2 is an ideal data source and testbed for future emission models of tropical rainforests, and it is the only enclosed forest where a $^{13}\text{CO}_2$ labelling experiment can be performed.

We now add the following text to the methods section to underline the suitability of the Biosphere 2 as a means of testing emission models of the future,

“The Biosphere 2 is therefore an ideal data source and testbed for future emission models of tropical rainforests and similar facilities for other forest ecosystems are necessary.”

F. Suggested improvements: experiments, data for possible revision:

The data on soil uptake would be needed for a proper understanding of the bidirectional exchange of monoterpenes.

We did make soil VOC flux measurements throughout this experiment at multiple locations in the Biosphere 2. The VOC emissions from/to the soil is the focus of a separate publication. The monoterpenes, which are the focus here, were only taken up during the experiment as shown in the Extended data figure 3 (Extended data figure 2 the original submission), apart from in the aftermath

of the first post-drought rain. This figure has been improved to show the soil flux of some selected monoterpenes and the sum of the total monoterpenes instead of the % exchange. The new figure is shown below.

Were there any isotopic measurements done from the soil chambers?

Yes, but since this is a follow on from the previous question, it may no longer be applicable in light of the previous answer that the monoterpenes were only taken up by the soil throughout the drying period. However, we did indeed periodically perform ^{13}C -pyruvate additions to the soil to evaluate how carbon was taken up and emitted by the soil. However, since the monoterpenes were only taken up by the soil instead of being emitted, it is not possible to evaluate the ^{13}C enrichment of monoterpenes emitted from the soil.

How did the enantiomeric fractionation of VOC uptake in the soil change and did that affect the results (especially as drought is a major disturbance for the soil microorganisms at least partially responsible for the uptake)?

Yes, this is an important point and perhaps we did not emphasize the importance of the soil flux data shown in extended data figure 3. These show that monoterpenes were taken up by the soil at a consistent rate throughout the drought period so that the percentage of uptake remained largely consistent.

In the revised manuscript on lines 128-130 it says:

“It should be noted that fluxes of monoterpenes from the soil did not affect these enantiomeric ratios as samples taken periodically throughout the experiment showed that the soil maintained a modest steady uptake of enantiomeric monoterpenes throughout (Ext. Data Fig. 3).”

We think this text, along with extended data figure 3, already make it clear that for the most part, soil did not affect the enantiomeric ratio of the monoterpenes. Nevertheless, to be sure we now add the sentence “Therefore, soil uptake did not drive the enantiomeric fractionation observed.”

Data on changes in VPD instead or in addition of the Atmospheric Water Potential would have been needed.

Atmospheric water potential data shown in figure 4 has now been replaced with VPD data as requested by the reviewer.

No analysis on the temperature impacts is included, that would be important for understanding the environmental drivers better.

This comment appears to be a repeat of the previous one related to temperature, which we hope has now been answered by the inclusion of emission model predictions as a function of temperature compared with measured data.

H. Clarity and context: lucidity of abstract/summary, appropriateness of abstract, introduction and conclusions:

The ms is written basically in good language, some typos are remaining which can easily be

corrected. Abstract is clear and summarizes the main results; however the indication of the results for atmospheric chemistry without any photochemistry data is not appropriate.

Regarding the implications for atmospheric chemistry, respectfully, we must disagree. Uniquely in this controlled environment, with atmospheric chemistry excluded and meteorological factors controlled, is the true ecosystem emission response to drought revealed. Since this differs from that predicted by current models it is reasonable to consider the potential implications of these findings for the atmosphere.

The underlying assumption is that the Biosphere 2 mesocosm realistically represents a rainforest ecosystem. The main difference is that because of the absence of atmospheric oxidation, isoprene concentrations rise to levels more than an order of magnitude higher than ambient data since the main Biosphere 2 sink for isoprene is the soil.

We make this assumption clear by inserting the following sentence:

“It should be noted that these conclusions are based on the assumption that the B2-TRF does represent the characteristic drought response of real-world tropical rainforests in the absence of atmospheric chemistry.”

Furthermore, the main point of the manuscript is that chiral emissions from plants respond differently to drought and are differentially enriched with ¹³C during atmospheric labelling which points to different underlying emission mechanisms. The presence or absence of atmospheric chemistry would make no difference to this point since enantiomers have the same rates of reaction with atmospheric oxidants.

With regard to atmospheric impacts, the shift in the time of the emission is particularly important as this will impact processes related to the formation of cloud condensation nuclei. This is because towards afternoon there is a shift in the partitioning between evaporation and sensible heat flux that favours the latter. More sensible heat flux means more turbulence in the afternoon which would facilitate vertical transport of the later emitted species to cooler more oxidative regions.

We now make this point more clearly by adding the following text,

“The shift in the timing of the emission is particularly important, as this will affect processes related to the formation of cloud condensation nuclei. Towards afternoon, there is a shift in the partitioning between evaporation and sensible heat flux that favors the latter. More sensible heat flux enhances turbulence in the afternoon, which would facilitate vertical transport of the later emitted species to cooler more oxidative regions.”

Referee #2 (Remarks to the Author):

The manuscript by Byron et al. describes observational data on emissions of chiral monoterpenes from the Biosphere II. While the emission of chiral monoterpenes has long been recognized, the reported findings are novel and highlight the need to improve BVOC emission models. Abstract and summary are well written and summarize the findings adequately. The impact of drought plays a major role in modulating BVOC emissions, and as such the presented results are an important piece

of information still missing when modeling monoterpene emissions. Data and methodology are adequately described, but there are some discrepancies with the accompanying paper that should be explained. Also, more detailed information on species distribution would be beneficial in order to better understand the ecological significance of the findings for a rainforest growing in the real world.

We thank the reviewer for recognizing the novelty and importance of this work. The reviewer's points are all addressed in detail below.

It is mentioned that 95 tropical species are grown in B2. What is the relative distribution of biomass?

In response to the reviewer's question, the following text has been added to the methods section between lines 11 - 16,

"The majority (~85%) of the biomass is in the large (DBH>15cm) trees, ~10% in the understory trees and ~5% in the understory herbaceous species (including *Musa* sp., *Alpinia* sp, *Hedychium* sp., and *Zingiber* sp., planted along the walls). *Clitoria fairchildiana* dominates the canopy (~33%). *Pterocarpus indicus* and *Hibiscus tilliceous* each take up ~15 and 10 percent of the canopy. All other tree species 5% or less each."

What is known about monoterpene emissions from these species (ie. how many of the species actually emit MT)? Can the results be distilled to a couple of high MT emitting species or are they more representative for the entire ecosystem?

This is a good point, and one that we did not make clearly enough. By measuring at 13 m in the center of the Biosphere 2 airspace, which was strongly mixed by fans, we capture the overall ecosystem response rather than a single species. The underlying assumption is that emissions by the vegetation in the Biosphere 2 represent those of the real-world rainforest. This is supported by the ratio of isoprene to monoterpenes which compares well to that measured in Amazonia when corrected for the lack of OH chemistry. Several plant species present were monitored in the laboratory prior to the experiment and found to only emit isoprene (e.g., *Pachira aquatica*) (Taylor et al. 2018). However, some of the main species in the Biosphere 2 emit both isoprene and monoterpenes and they respond differently to the drought. In order to make this clear we have now added to the extended data, emission profiles for the total monoterpenes measured from 4 *C. fairchildiana* trees and 4 *Piper* sp. plants (from cuvettes) at certain points throughout the measurement time. As can be seen from the data the responses are different and what we are measuring is therefore the net response of this ecosystem.

Accordingly, we now add the following text:

"Furthermore, the air was strongly mixed with fans, resulting in the measurement of the total ecosystem response rather than a single species. The periodic measurements of monoterpene emissions from 4 *C. fairchildiana* trees and 4 *Piper* sp. plants (from cuvettes) are provided in **Ext. Data Fig. 4**. The individual plant responses were buffered in atmospheric measurements, showing that the atmospheric measurements were the net response of the ecosystem."

This also relates to the connected interpretation of Fig3 and associated modelling exercises. Why can it be assumed that chiral monoterpenes are exclusively emitted internally mixed (i.e. de novo

and stored (only) from the same plant)? Couldn't it just be that different species emit different chiral monoterpenes, and the actual effect seen is that some species see largely different responses to drought stress than other species?

We agree with the reviewer that different plant species do emit different chiral monoterpenes and have different responses to drought. This is also now highlighted by the addition of the emission data from the 4 *C. fairchildiana* trees (canopy) and 4 *Piper* sp. (understory) plants. That is why a valid approach to understanding how drought affects monoterpene emissions is to focus on the combined ecosystem response rather than measuring each species independently since different plant species have been previously shown to have different responses to drought stress. Studying the combined ecosystem response to drought is a more effective means of determining how the overlying atmosphere will be affected.

We insert the following line to clarify this point, which we have already mentioned in response to a previous comment,

“The periodic measurements of monoterpene emissions from 4 *C. fairchildiana* trees and 4 *Piper* sp. plants (from cuvettes) are provided in Ext. Data Fig. 4. Since the individual plant responses are largely different to what was measured in the atmosphere, the atmospheric measurements were the net response of the ecosystem.”

Deciphering the relative distribution of MT emitting species and their individual response to the drought would be important to support the underlying conclusion of this study. During severe drought stress one would also expect other VOCs emitted (e.g. green leaf volatiles, or compounds from the shikimate pathway)? Is there any additional evidence from other ‘marker species’, and how do their emission patterns during the drought correspond to the changing chiral pattern of MT emissions?

Since a quadrupole mass spectrometer operating in selected ion mode was used, only the targeted isoprene and the chiral monoterpenes were measured throughout the entire measurement campaign with the on-line GC-MS. Moreover, we were only able to measure a limited number of the abundant plant species individually with cartridges and these data are now included to highlight the differing responses (see point above). The main focus of this paper is chiral monoterpenes, however, to address the reviewer’s point, we now make reference to the Werner paper, which shows measurements of the green leaf volatile hexanal peaking late in the severe drought and insert the following text.

“At the end of the severe drought when emissions of the chiral monoterpenes begin to decrease, the stress marker hexanal was observed to increase indicating leaf damage (Werner et al. 2021)”

What was the exchange rate of air during the experiment? Some details are still unclear concerning sources and sinks of BVOC. In the accompanying paper the authors state that “Due to the UV-screening glass of Biosphere 2, atmospheric oxidation effects are negligible and atmospheric VOC concentrations reflect the interplay between vegetation emissions and net exchange rates with soil”. This seems too simplified to me – wouldn't the exchange rate through the air handling system also be a major sink due to mixing with ‘fresh outside air’, or how was this accounted for (e.g. using SF6

data) and/or factored in the data analysis? It would be good to know how fast the air was exchanged, as it could serve to determine actual net exchange rates, from which net emissions per unit biomass could be calculated. This could at least help comparing B2 emissions with those obtained in real-world ecosystems in the tropics. How similar or different are these? Putting results more in context of the real world would help strengthen the ecological significance of the conducted experiments.

The accompanying paper, which is now published (Werner et al. Science et al. 2021) gives a comprehensive description of the experiment and the facility. The air exchange rate is indeed accounted for by using SF₆ measurement data as is already described on lines 48-52 in the methods section. The exchange of the rainforest air with the fresh outside air was corrected for with the equation on line 50 in the methods section. This equation uses the exchange rate of measured SF₆ to correct the atmospheric concentration of monoterpenes. The plot of SF₆ as a function of time has now been provided in the extended data to make this point clearer. We have also provided more detail on the methodology of how SF₆ was put into the atmosphere of the Biosphere 2 and measured with the following text:

“To measure the air-exchange rate, 25 to 30 ml SF₆ was injected into the rainforest, thereby generating a concentration of ~1 ppb (~125 times background air at ~8 ppt). SF₆ sampling took place next to the instrument laboratory using a single, filtered inlet connected to ¼” OD Teflon tubing. The concentration of SF₆ was measured using an SRI Greenhouse Gas GC (SRI Instruments, Torrance, CA, USA) with an automated sample loop of 1 ml using an ECD detector at 350 °C. A Hayesep D column at 65 °C was used to separate the SF₆ in the sample and the UHP N₂ carrier stream from N₂O. Samples were collected and analysed every 2.5 minutes. The exponential decay of the SF₆ concentration in the Biosphere 2 rainforest was used to calculate the exchange rate and was reported as % per hour.”

We appreciate the suggestion about determining net emissions per unit biomass and we agree that this helps with putting the results in the context of the real world. Therefore, we have calculated the isoprene and monoterpene fluxes relative to the land surface area and tree biomass carbon. The fluxes were calculated based on the change of moles in the system, the moles exchanged with the flow through, and the soil uptake rate calculated based on the rate of change in moles during early night time hours relative to the gas concentration (assuming that the gas concentration has a strong

influence on the uptake rate (Pegoraro et al. 2005). This table is now included in the extended data (Extended data table 3) and is shown below.

Drought Stage	Isoprene gC ha ⁻¹ d ⁻¹	Isoprene mgC kgC ⁻¹ d ⁻¹	(-)- α -pinine mgC ha ⁻¹ d ⁻¹	(-)- α -pinine mgC kgC ⁻¹ d ⁻¹	(+)- α -pinine mgC ha ⁻¹ d ⁻¹	(+)- α -pinine mgC kgC ⁻¹ d ⁻¹
Pre-drought	0.52±0.26	8.2±4.0	0.79±0.50	12.5±7.9	0.04±0.55	0.59±0.80
Early drought	1.44±0.65	22.6±10.3	3.24±1.34	51.1±21.1	0.40±0.43	6.35±6.71
Severe drought	0.38±0.43	6.1±6.7	3.82±1.09	60.1±17.1	0.58±0.36	9.15±5.7
Deep rewet	0.29±0.06	4.6±1.0	4.87±0.57	76.7±8.9	0.31±0.02	4.96±0.34
Recovery	0.54±0.45	8.6±7.1	5.22±3.44	82.1±54.1	0.19±0.18	2.96±2.86
Drought Stage	(-)- β -pinine mgC ha ⁻¹ d ⁻¹	(-)- β -pinine mgC kgC ⁻¹ d ⁻¹	(-)-limonine mgC ha ⁻¹ d ⁻¹	(-)-limonine mgC kgC ⁻¹ d ⁻¹	(+)-limonine mgC ha ⁻¹ d ⁻¹	(+)-limonine mgC kgC ⁻¹ d ⁻¹
Pre-drought	0.08±0.07	1.2±1.2	0.08±0.10	0.12±1.61	0.02±0.01	0.33±0.21
Early drought	0.27±0.14	4.3±2.2	0.15±0.09	2.35±1.37	0.01±0.01	0.10±0.17
Severe drought	1.60±1.71	25.1±27.0	0.19±0.14	2.94±2.14	0.03±0.04	0.39±0.65
Deep rewet	1.86±2.24	29.3±35.3	0.12±0.08	1.86±1.34	0.04±0.01	0.69±0.03
Recovery	0.94±0.22	14.8±3.5	0.01±0.03	0.18±0.47	0.04±0.06	0.54±0.88
Drought Stage	(-)-camphene mgC ha ⁻¹ d ⁻¹	(-)-camphene mgC kgC ⁻¹ d ⁻¹	(+)-camphene mgC ha ⁻¹ d ⁻¹	(+)-camphene mgC kgC ⁻¹ d ⁻¹	γ -terpinene mgC ha ⁻¹ d ⁻¹	γ -terpinene mgC kgC ⁻¹ d ⁻¹
Pre-drought	-0.002±0.003	-0.031±0.046	-0.005±0.005	-0.083±0.082	0.041±0.028	0.639±0.448
Early drought	0.004±0.004	0.068±0.057	0.017±0.007	0.267±0.116	0.090±0.049	1.412±0.778
Severe drought	0.010±0.003	0.154±0.051	0.056±0.034	0.876±0.535	0.480±0.738	7.551±11.615
Deep rewet	0.001±0.001	0.013±0.015	0.034±0.007	0.530±0.104	-0.040±0.138	-0.637±2.174
Recovery	0.001±0.006	0.013±0.101	0.029±0.040	0.464±0.637	0.012±0.006	0.184±0.098

Isoprene concentrations for example reached levels up to 500 ppbv. Those of total monoterpenes only 2 ppbv. This is only about 0.4%. Typical MT emissions over tropical forests are quite a bit higher relative to isoprene. Is there an explanation based on MT emitting species in the B2?

It is very likely that drought affects the ratio of monoterpenes to isoprene and we are lacking real-world values comparable to 20 days of drought.

However, if we instead take the pre-drought values of 100 ppbv of isoprene and 1 ppbv of monoterpenes then this gives a percentage of 1%. A. M. Yáñez-Serrano et al. 2020 has reported understory isoprene mixing ratios ~ 5 ppb, and for total monoterpenes, as low as 0.1 ppb. This equates to a percentage of monoterpenes to isoprene of $\sim 2\%$. The reaction rates for isoprene and monoterpenes with OH is approximately $K_{\text{isoprene+OH}} = 1 \times 10^{-10}$ and $K_{\alpha\text{-pinene+OH}} = 5 \times 10^{-11}$, respectively (Atkinson, R., 1986). $K_{\text{isoprene+OH}} / K_{\alpha\text{-pinene+OH}} \sim 2$. Thus, if atmospheric chemistry is removed from a system, the ratio of isoprene to monoterpenes should decrease by a factor ~ 2 which is indeed what we see.

Of course, a wide range of isoprene and monoterpene concentration values have been measured in the Amazon rainforest as shown in A. M. Yáñez-Serrano et al. 2020 and this is most likely dependent on season, measurement location and the plant species in the vicinity of the sampling inlet. Different values will give different percentages but nonetheless, the absence of atmospheric chemistry will increase the isoprene concentration greater than the monoterpene concentration, thereby lowering the monoterpene to isoprene ratio. Additionally, it is also reasonable to assume that the low isoprene to monoterpene ratio in the Biosphere 2 reflects the fact that over half (~56%) of the Biosphere 2 vegetation are strong isoprene emitters (as is the case in most real tropical forests) (Taylor et al., 2018). We have conducted sampling experiments in the past on other tropical plant species namely *Pachira* and *Ficus* plant species (species that are found in abundance in the Biosphere 2) and found almost no emission of monoterpenes but very high emissions of isoprene.

This point is addressed by the text inserted to explain the absence of chemistry.

“No significant OH oxidation chemistry can occur in the B2-TRF since glass does not transmit light at wavelengths that generate OH, and ozone within fresh incoming air is lost to the surfaces of large air handling units. The absence of any significant photochemistry is reflected in the ratio of isoprene to its oxidation products (which is 100 times richer in isoprene than in typical Amazon rainforest measurements) and in the isoprene to monoterpene ratio (which favors isoprene in the Biosphere 2 by a factor of ~3 due to its faster OH reaction coefficient). Ozone was also measured post-campaign within the B2-TRF and found to be at very low concentrations, on average 1.1 ± 0.7 ppb whilst outside the Biosphere 2, the air was found to contain an average ozone concentration of 49.2 ± 1.2 ppb.”

Also, in the accompanying Science paper quite a bit lower isoprene concentrations were reported for the same (?) period (e.g. ~500 ppbv on day 300 in this paper vs. ~150 ppbv on day 300 in the paper submitted to Science). Or was this a different year – if so what were the differences?

In the Science paper for consistency with accompanying soil and cuvette flux data we used PTR-ToF-MS data to represent the isoprene measurement. Here the focus is on enantiomeric monoterpenes which can only be measured by GC-MS techniques as pre-separation is required. Therefore, again for consistency we chose to use the isoprene measured by the same GC-MS instrument. Although broadly similar in the temporal behavior which is important for this work, the isoprene traces from both systems diverged somewhat in the early drought period (PTR-ToF-MS was lower). Despite rigorous investigation of both systems no cause for the discrepancy could be found despite the inlets being closely located to each other. The only remaining plausible explanation was that the differing inlet flow rates had caused different locally influenced air to be sampled so we chose to use the isoprene measured by the same device in exactly the same location for this analysis. For transparency, this has now been pointed out in the methods section. Since we only use the temporal behavior of isoprene as an indicator of the general behavior of the de novo emission signal the differences in isoprene concentrations between the instruments are not important in this context and the same conclusion can be drawn using the other PTR-MS based dataset. We have included the following text in methods section,

“Atmospheric isoprene was also measured from the same position with a different sampling inlet and line with a PTR-TOF-MS. Even though the temporal behavior was broadly similar between both datasets, the measured concentrations from both datasets diverged temporarily in the early drought period (PTR-TOF-MS measured lower concentrations). Despite rigorous testing of both systems no cause for discrepancy could be found, therefore the only plausible explanation was determined to be that differing inlet flow rates caused different locally influenced air to be sampled.”

Minor comments:

Figure 2 shows some sort of model smoothed BVOC concentrations, in my opinion it would be beneficial to add actually measured concentrations in this figure as well (e.g. aggregated daily means or similar). What is the shaded modelled line (measured uncertainty or real daily variability in the B2)?

In section 4.3.4 of the manuscript, it is explained that the BVOC concentrations were processed through a Savitzky-Goulay filter to keep the long-term pattern whilst removing short-term fluctuations. A further processing through the MATLAB smooth filter was then applied to suppress the noise on the resulting line. Showing aggregated daily means, as suggested here by the reviewer, results in the short-term fluctuations dominating the signal pattern and the long-term pattern becoming less obvious, which offers no advantage to the reader in helping them spot long-term differences in the responses of the BVOCs to progressive drought and rewetting.

Therefore, we think that it is best to keep the current figure as it is but we have put a new figure in the extended data showing the aggregated daily means, shown below (Extended Data Figure 15).

The shaded region around the line in figure 2 represents the measurement uncertainty as is already explained in the figure caption. By this we mean the calculated uncertainty of the measurement which has been propagated through the Savitzky-Goulay and smoothing filters with the rest of the data. This can be viewed as an uncertainty on the processed measurement. This has now been made more clear in the figure caption by adding “calculated” to the penultimate sentence.

Figure 2 caption: “**Figure 2.** Long-term total monoterpenes (MTs) and isoprene trends throughout the experiment show strongly differing trends for monoterpene enantiomers, especially during daylight hours. Monoterpene and isoprene data are divided into 5 stages, indicated by the grey dashed lines: pre-drought (PD), early drought (ED), severe drought (SD), deep-water rewet (DRW) and rain rewet (RRW). The timing of the $^{13}\text{CO}_2$ pulses is indicated by the black (---) lines. **a.** Daytime isoprene and total monoterpene volume mixing ratios (VMR). The shaded region around the line represents the absolute measurement uncertainty. **b.** Average daytime (-) and (+) alpha-pinene and other monoterpenes. **c.** Average nighttime (-) and (+) alpha-pinene and other monoterpenes. **d.** soil moisture (SM) and relative humidity (RH). Note different scales for enantiomers. **e.** Pie charts showing the daytime composition of the enantiomeric monoterpenes during each stage. Note different scales for enantiomers. The shaded region around the line represents the calculated measurement uncertainty. Other monoterpenes includes (-)-camphene, (+)-camphene, (-)-limonene, (+)-limonene and γ -terpinene.”

Referee #3 (Remarks to the Author):

Ambient air concentrations of monoterpenes were monitored throughout a drought experiment in a tropical forest ecosystem enclosure. Changing relative contribution of diverse monoterpenes to whole system total monoterpene emissions during the experiment were observed as well as different labeling intensity of enantiomers of α -pinene and different shifts in diel cycles of diverse enantiomers in response to the drought treatment. The results evidence the role of differences in residence times of diverse emission sources.

I suggest to change the title of the paper, as the results evidence the relevance of emissions from sources of different residence times rather than reveal mechanisms.

This is an interesting and thought-provoking comment. If indeed the emissions were the result of a single enzymatic process and the differences observed were then merely the result of delayed versus instantaneous emission then we would agree. However, there appears to be multiple

enzymatic processes at play, coupled with aqueous and non-aqueous storage pools, even in the case of the simplified model explanation presented here. We therefore feel that the word mechanism more accurately expresses the somewhat more complex situation that is shown here. It encapsulates all of the processes leading up to emission (synthesis, storage, stomatal opening) better than residence time alone.

Several bits of additional information needed to fully understand the experiment are communicated with the companion paper only, as e.g. location of Biosphere 2, height of the enclosure, non-use of additional lighting, time course of $\delta^{13}\text{C}$ of the atmosphere, operations applied to impose early and severe drought. They should be added to the text and extended data.

This point has been raised by all reviewers and so we now add to the supplementary section the following text which describes in more detail the distribution of vegetation, dimensions of the enclosure, non-use of additional lighting. We also refer the reader to the companion paper Werner et al., which describes this in more detail.

“and is located near Tucson, AZ, USA”

“Additional lighting was not used within the tropical rainforest.”

“To ensure timely progress of drought, manipulation of the ecosystem moisture began with turning off the aesthetic water features (waterfall, pond, stream) before the experiment on 31st May 2019.”

“Relative humidity was actively reduced using a large air handler unit during the severe drought (1st November 2019 – 2nd December 2019). Air was first cooled to create condensation, and then re-heated to maintain temperature. To enhance drought conditions, a persistent water table in the isolated drainage basin of the Varzea sub habitat was drained throughout the severe drought period.”

“The rainforest was again watered from above using the sprinkler system and ~35,000 l of water, at 11:00 on 12th December 2019 and ~36,000 l of water at 11:00 on 19th December 2019. The rainforest was then watered at 00:00 every 2 days afterwards, adding 20,000 l of water to the rainforest over a 4.5 hour period.”

The statement in the Methods section line 52 likely is true. However, the basis of this assumption should be explicated with one phrase and if possible a reference added.

The statement that the reviewer refers to on line 52 is “The incoming VMR of all VOC’s were assumed negligible”. This has now been changed to say “and incoming VOC VMRs were assumed negligible.” The desertic landscape around the Biosphere 2 site is largely absent of any foliage or vegetation besides rock and scrub, so despite the absence of a reference we think that it is very safe to assume that none of the compounds that are being presented in this manuscript could have come from an external source outside of the Biosphere 2.

Sampling was performed at the height where the canopy is most dense. Can information on within system gradients (CO_2 , water vapor) be supplied?

We have now added a table to the extended data (extended data table 4 and 5) which shows the average temperature, CO₂ and water vapour for the understory and canopy for each phase of drought.

Total day and night average

Drought Stage	CO ₂ ppm	CO ₂ ppm	VPD	VPD	T °C	T °C
	understory	canopy	understory	canopy	understory	canopy
Pre-drought	472±20	456±19	0.25±0.01	0.83±0.13	22.4±0.6	26.8±0.6
Early drought	445±8	429±7	0.81±0.22	1.39±0.21	24.8±1.4	27.8±1.1
Severe drought	458±6	440±6	1.39±0.24	1.75±0.22	24.6±0.7	26.5±1.0
Deep rewet	451±13	433±13	1.19±0.09	1.45±0.10	24.1±0.5	25.7±0.4
Recovery	453±10	433±11	1.10±0.16	1.41±0.20	24.8±0.5	26.8±0.7

Day time average only

Drought Stage	CO ₂ ppm	CO ₂ ppm	VPD	VPD	T °C	T °C
	understory	canopy	understory	canopy	understory	canopy
Pre-drought	388±22	364±21	0.27±0.02	1.42±0.24	23.1±0.8	30.8±0.8
Early drought	402±11	380±11	0.98±0.14	2.22±0.16	26.8±0.8	32.4±0.9
Severe drought	434±5	411±6	1.61±0.18	2.30±0.34	26.6±1.0	29.8±1.6
Deep rewet	417±3	394±4	1.11±0.14	1.78±0.21	26.0±1.0	29.5±2.6
Recovery	405±16	380±18	1.18±0.12	1.62±0.29	26.8±0.7	29.3±1.3

Can supporting information on the absence of potential reactants for monoterpenes in the enclosure air be given?

This is a point that is also raised by reviewer 1 and it is also addressed in the general responses. We recognize that the Biosphere 2 will be a new environment for many readers of this paper and therefore we have provided further explanation of why atmospheric chemistry is negligible in the mesocosm.

Atmospheric chemistry from OH, O₃ or NO₃ was not significant in the biosphere. This was because large air handling units largely remove ozone from the rainforest air due to surface losses, and glass does not transmit the UV wavelengths responsible for OH radical generation. This is also supported

by the ratio of isoprene to its oxidation products, which is around 100 times higher than seen in outdoor rainforests. Subsequent ozone measurements showed an average ozone concentration of 1.1 ± 0.7 ppb inside the rainforest, whilst outside the average ozone concentration was found to be 49.2 ± 1.2 ppb. Furthermore, the ratio of isoprene to monoterpenes is similarly enhanced according to the relative rate of OH radicals with both species. Therefore, it appears that the Biosphere 2 does not realistically represent rainforest emissions in the absence of atmospheric chemistry.

In order to substantiate this point we now add the following text to the manuscript,

“No significant OH oxidation chemistry can occur in the B2-TRF since glass does not transmit light at wavelengths that generate OH, and ozone within fresh incoming air is lost to the surfaces of large air handling units. The absence of any significant photochemistry is reflected in the ratio of isoprene to its oxidation products (which is 100 times richer in isoprene than in typical Amazon rainforest measurements) and in the isoprene to monoterpene ratio (which favors isoprene in the Biosphere 2 by a factor of ~ 3 due to its faster OH reaction coefficient). Ozone was also measured post-campaign within the B2-TRF and found to be at very low concentrations, on average 1.1 ± 0.7 ppb whilst outside the Biosphere 2, the air was found to contain an average ozone concentration of 49.2 ± 1.2 ppb.”

Show the time course of air $\delta^{13}\text{C}$ measurements during and after the labeling pulses with a figure in the extended data (Methods section lines 147-148). The information available with the companion paper indicates that ^{13}C label returned to the air of the enclosure in significant amounts for several days due to respiration of diverse ecosystem components. Thus, though the enclosure was ventilated at the end of the pulses, photosynthesis used weakly labeled source CO_2 for several days.

Figure 3 has now been modified/extended to show the enrichment of ^{13}C on the two subsequent days after each ^{13}C labelling pulse. This means that figure 3 now shows that there was not a significant ^{13}C enrichment in (-)- α -pinene in the days after the labelling pulse. In regards to this, the sentence on lines 167-170 of the revised manuscript has been changed to say “Atmospheric samples taken post-pulse show that, on average, the baseline $\epsilon^{13}\text{C}$ values of (-)- α -pinene did decline to pre-pulse values. This shows that (-)- α -pinene emissions are predominately *de novo* but, it should not be completely ruled out that a small fraction also enters the storage pools from which it is emitted after the labeled CO_2 is flushed from the TRF.”

Since we only have a small collection of atmospheric samples from the labelling pulse and the days thereafter, and not highly time-resolved data, we cannot show the time course of ^{13}C measurements. (Also in the companion paper (Werner et al. 2021) ^{13}C -labelled VOCs were only detected on the day of the ^{13}C -pulse in significant amount, thereafter they were below the detection limits.)

Moreover, the atmosphere was only 20-30 per mil enriched in ^{13}C relative to pre-pulse conditions on the day after the labelling pulse, in contrast to $\sim 1000/\sim 2000$ per mil on the day of the labelling pulse. Furthermore, two days after the pulse, ^{13}C enrichment of the atmosphere was still detectable but only a couple of per mil above the baseline, which is not likely to impact the monoterpene $\epsilon^{13}\text{C}$ values.

There are inconsistencies between the dates for transition from early drought to severe drought and end of severe drought between figure 2 and figure 2 of the companion paper.

We thank the reviewer for pointing this out and we have now changed the plots so that the papers are consistent with each other. The pie charts have also been changed due to the shifting of the exact drought phase boundaries.

In addition at line 103 peaks in monoterpene mixing ratio after 33 days and after 66 days are mentioned, which according to figure 2 rather occur at 23 days and 56 days, respectively. The description of the sensitivity analyses in lines 197-198 for the quantification of (single) peaks in the GC-IRMS chromatograms shown with Fig. 6 in the extension data does not reflect what has been shown with the figure.

We again thank the reviewer for pointing out the inconsistency between the figure and the text regarding the peaks in the monoterpene mixing ratio, we have now corrected the manuscript.

With respect, we disagree with the reviewer's comment that the description of the sensitivity analysis in lines 197-198 does not reflect what was shown in extended data figure 6 (now extended figure 11). The figure clearly shows that by integrating the first peak alone or in combination with the subsequent co-eluting peaks, there is a clear enrichment of 13C. But when the second peak is

integrated alone (i.e. trough to trough), there is no enrichment, which would also be the same for the subsequent peaks. Therefore, the first peak (which is (-)-alpha-pinene) must be the peak that is enriched in 13C.

Suggestions for refocusing the paper:

The correlation of α -pinene enantiomer concentrations with other MT (line 26) is not communicated with the paper.

We thank the reviewer for pointing this out and we made the following changes: we have now made tables showing the Pearson's correlation coefficients of all of the monoterpenes and added it to the extended data. The tables are shown below.

The following text has also been added to the manuscript: “During pre-drought, (-)- α -pinene concentration correlated better with the concentration of (-)- β -pinene, (-)-limonene, (+)-limonene, and (+)-camphene than it did with the concentration of (+)- α -pinene (Ext. Data Fig 1). Inversely, during severe drought, the concentration of (-)- α -pinene correlated better with (+)- α -pinene than with any other measured compound.”

Though the physical properties of α -pinene enantiomers do not differ much (Lines 34-35), a recent study showed differences in the physical properties of derived dimers: Bellcross A., Bé A.G., Geiger F.M. & Thomson R.J. (2021) Molecular Chirality and Cloud Activation Potentials of Dimeric α -Pinene Oxidation Products. Journal of the American Chemical Society.

We thank the reviewer for making us aware of this extremely interesting work. We did not consider it in our paper since it was published in October 2021, 2-3 months after we submitted this manuscript. Our work to date has been confined to the original chiral emissions and these do not show any physio-chemical differences between enantiomers for physical data or reaction coefficients. We have also tested their uptake rates onto Amazonian aerosol (see Zannoni et al. Nature Earth Communications 2020) but again found no discrimination between the monoterpene enantiomers. Therefore, in that sense the point made in the manuscript is correct that the physical properties are equal for the precursor monoterpenes.

However, the aforementioned work suggests that photochemical dimer products of the oxidized monoterpenes do indeed show different physical behaviours. In particular that mixed enantiomer product dimers are more hydrophobic. We therefore now make the general point based on this paper that the combinations of chiral photochemical products can exert different physico-chemical effects in the atmosphere.

Accordingly, we insert the following text in the concluding section

“While monoterpene enantiomers exhibit no difference in physical properties, in oxidation rates by OH or O₃, or in uptake rates to typical amazon forest aerosol samples (Zannoni et al. 2020), recent work has shown that dimeric photochemical product combinations do indeed have different hydrophobicities (Bellcross et al. 2021).”

I suggest to add the description of the dynamics of the ratio (parallel rise and earlier peak in isoprene) through the early drought period to lines 129-132.

The following text has been added to the manuscript:

“During early drought, isoprene and total monoterpenes rose in parallel, with isoprene peaking earlier” and “..., whereas the total monoterpenes continued to rise again in severe drought”

The text on lines 177-180 provides a misleading description. The conclusion can be drawn for the α -pinene only as parameterization of the light and temperature response functions does differ between different monoterpenes.

To avoid misleading the reader, we have now changed the text so that “all monoterpenes” is replaced with “alpha-pinene” so the manuscript now says:

“Current atmospheric models would predict alpha-pinene emissions as a function of light and temperature and would erroneously place peak alpha-pinene emission midway between the real peaks and would be unable to reproduce the drought induced changes revealed by enantiomers.”

More information on the post-pulse dynamics of $\epsilon^{13}C$, only addressed with a summary statement (lines 149-150), should be given, as the storage issue is a central topic of the paper. In addition add information on the time frame within which post-pulse samples were taken to line 157 in the methods section.

I suggest to more deeply explore the night time and long-term emission effects of labeling recognized as largely unknown terrain by Niinemets & Reichstein.

In light of the new Figure 3, lines 149-152 have now been changed since it is no longer shown that the $\epsilon^{13}C$ decline rapidly to pre-pulse values. The text now says “Atmospheric samples taken post-pulse show that, on average, the baseline $\epsilon^{13}C$ values of (-)- α -pinene did decline back to pre-pulse values. This shows that (-)- α -pinene emissions are predominately de novo but, it should be not be completely ruled out that a small fraction also enters the storage pools from which it is emitted after the labelled CO₂ is flushed from the TRF.”

We agree that more information on the post-pulse dynamics should be provided. We have changed figure 3 so that the 2 days directly after the labelling day are now shown, as shown below.

We also thank the reviewer for their suggestion to explore the night-time and long-term emission effects of labelling. Unfortunately, we did not conduct atmospheric sampling during the atmospheric labelling periods during the night-time and samples were only collected for two days after the labelling pulse. Therefore, a deeper exploration of the night time and long-term emission effects of labelling is not possible.

The conclusion that the results „show“ (L229) that changes in fractional contribution of monoterpenes can be used to gauge drought in tropical ecosystems is a bit premature. It rather is a suggested hypothesis that requires further testing.

On reflection we agree that we should suggest this method of gauging drought on the basis of these findings, and state that it requires further testing. Accordingly, the text has been modified so that instead of “show” it says “suggest”:

“These results suggest that the degree of drought stress in a tropical rainforest can be gauged by either the afternoon-to-morning ratios of (-)- α -pinene or by the fractional contribution of (-)- β -pinene to the sum of monoterpenes, which almost tripled from pre- to severe drought. However, further testing of this hypothesis under real world conditions is required.”

The conclusion on identification of sources and changes in carbon cycling (L234-236) is ambiguous. Rather the measurements indicated the existence of source pools with different half-life times and changes in their relative contribution to emissions in response to drought. However, the identity of

the pools within the complex ecosystem comprising many different species, differentially exposed to light and VPD during the day, potential translocation and absorption by different plant organs can not be deduced from the presented results.

Our intention is to indicate possible ways forward in revising current emission models by studying holistically the combined system of the rainforest. We show results on the ecosystem scale and therefore we identify the pools from where the majority of individual monoterpenes came from on this scale.

Finally the conclusion about the need to include enantiomerically resolved monoterpenes in large scale models (L240-242) is far to blunt. The relevance of this step has not been studied with the current paper. The general request for "accurately assessing" has no scientific sense. No model is a one-to-one digital representation of reality. The accuracy that can be achieved with the simplifying assumptions of model structures needs to be gauged against specific objectives, and for specific target values.

Our results reveal that, contrary to the underlying assumptions of emission models, the emissions of individual monoterpene enantiomers respond differently to drought, both in absolute rate and time of day. Our overall long-term goal is to understand the atmospheric impact of the tropical rainforest ecosystem drought response. If this goal is pursued on the basis of monoterpene data in which both enantiomers are combined, then only a blurred understanding and predictive capability is achievable. This is the fundamental revelation of this paper. However, the reviewer's comment does show that this point needs to be made more carefully. We have rephrased the text as follows,

On line 239, "accurately" has been removed and replaced with "faithfully".

Emission models require a fair number of experimental studies that provide results on emission factors in order to parameterize the emission factors (EF). The presentation of the updating of EF between two model versions in Messina et al. 2016 illustrates well the importance of this fact. Thus, prior to distinguishing between chiral forms the number of experimental results needs to be increased. This issue is further underlined by the differences between emissions simulated with the ORCHIDEE and MEGAN models shown in the same paper, indicating the relevance of other factors (e.g. number and distributions of plant functional types, scarcity of experimental information for parameterizing the light response) impacting the flux calculations.

We agree with the reviewer that many experimental studies need to be performed for the rigorous parametrization of emission factors. However, we do not agree that this should be done prior to distinguishing between chiral forms. What we have shown in this work is that chiral forms have different dependencies on light, temperature, and drought, and that separation of chiral forms can be achieved whilst measuring other terpenoids at no disadvantage. Thus, we are confused as to why we should wait to distinguish between chiral forms when we have shown the importance of enantiomeric separation and separation can already be achieved. We hope that this work makes it clear that future experimental studies need to separate and independently measure enantiomers so that chiral emission factors can be rigorously parametrized.

Furthermore, as shown by Niinemets and Reichstein 2002, further refinement of models requires to include storage pools as emission rates probably are rarely in steady state. However, they also recognized that including further detail into the models applied at large scales may not improve their performance. Temporal variation in emission of diverse compounds plays an important role. How important is the single factor of chiral forms?

We agree with the reviewer that current emission models need to be improved, which is perfectly normal as ongoing research reveals increasing levels of detail and complexity. The point that there are multiple areas within existing emission models that can be addressed and it is not yet clear how the relative impacts is well taken. Nonetheless, our point remains that fundamentally treating two independently controlled chemical species as one is flawed. This step is therefore at the core of all studies.

I suggest to focus more on discussion of the ecosystem components potentially involved in causing the observed changes in monoterpene emissions and progress made relative to the papers on monoterpene emission including enantiomers cited in the introduction instead of repeatedly making strong statements about consequences for larger scale modelling.

The first part of this point has been addressed in previous responses in which we now include data taken from cuvettes around individual plant species. The species analyzed show different responses confirming that the data presented here from the well-ventilated central space of the BIOSPHERE represent the net ecosystem response which is the key information sought.

The ultimate goal for atmospheric scientists is understand how environmental changes will change the abundance of atmospheric VOCs and how that will influence the climate response. Since it is not possible to faithfully evaluate how monoterpenes respond to environmental changes without distinguishing between enantiomers, we feel it is necessary to stress the point of how separating enantiomers can help to improve large scale modelling. The papers that include data on enantiomers (such as those cited in the introduction) did not measure enantiomers during progressive drought and recovery and therefore we find it inappropriate to use this work to explain the past studies suggested by the reviewer. Furthermore, other than Zannoni et al. 2020 and partially Williams et al. 2007, all the other publications that are cited in the introduction that distinguished between enantiomers were studies on boreal and Mediterranean plant species. Since the dominant alpha-pinene enantiomer has already been shown to be different for a boreal forest and tropical rainforest as shown in Williams et al. 2007, we feel that it is not appropriate to try to use a study on a tropical rainforest ecosystem to explain observations on boreal and Mediterranean ecosystems.

Suggestions for editing:

L17 why „but“

L20/21 instead of „without photochemistry“ better „in absence of UV light“

L23 emitted (-)- α -pinene mainly de novo \rightarrow emitted mainly de novo synthesized (-)- α -pinene

L24 (-)- α -pinene emissions \rightarrow the source of (-)- α -pinene emissions

L25/26 The α -pinene enantiomers each \rightarrow Mixing ratios of both α -pinene enantiomers

L60 compound \rightarrow compounds

L60-62 Phrase incomprehensible. What do you mean by intractable?

L66 delete „measuring“

L97 Is this an averaged over the whole periods?

L131 This means that → Thus

L157 Rather: „... (-)- α -pinene trans- β -ocimen and β -myrcene synthesized with photosynthetic assimilates produced during the labeling pulse were directly emitted ...“

L160-162 awkward wording, rephrase

L197 from the → from storage in the; emissions fluxes most likely have to transit the aqueous phase

L199 quantities → fractions

L224 when → with

Methods

L47 delete „be“

L176 the H₂O → of H₂O

These suggestions from the reviewer are well received and we have made changes to the manuscript accordingly.

Reviewer Reports on the First Revision:

Referees' comments:

Referee #1 (Remarks to the Author):

The manuscript has been revised and important with this, previously missing or inadequate information was provided. Further, as the accompanying paper (Werner et al) was now published, it provides a good reference for several background data which otherwise would have been somewhat speculative. Most importantly, the authors now justify the importance of their findings e.g., with the comparison of data with the widely used MEGAN model and show that during the growth the model predictions are poorly matching the data, likely due to complexity in biosynthesis and storage processes related to the enantiomeric distribution of monoterpenes, that changes during drought period.

Thus, I conclude that the ms is now from my perspective recommended for being accepted in Nature.

Referee #2 (Remarks to the Author):

The authors have addressed many concerns and improved the manuscript quite a bit. Overall the study presents an interesting aspect on the specific nature of certain BVOC emitted from tropical environments. The quantitative nature of the results will likely spur some debate in the analytical community, considering that isoprene concentrations (a compound that should be easy to measure quantitatively in such a setting) differed by a factor of up to ~ 4 between two complementary analytical techniques during the same campaign.

Referee #3 (Remarks to the Author):

The revised manuscript essentially addressed my comments and suggestion made during the first round of review and is substantially improved. The merit of the paper is to show the relevance of differences in the dynamics of release of terpene enantiomers from a tropical ecosystem.

I still disagree with the direct line drawn from these findings to model improvement. However, this disagreement is not relevant for the evaluation of the revised paper.

Several aspects remain to be studied in order to gauge the relevance of your findings for modelling. As mentioned in response to reviewer 1 the difference in peaks of light and temperature are less important in the real-world than within the enclosure which probably will reduce the error introduced by a pooled representation of the enantiomers. The comparison of measured concentration to MEGAN flux estimates Fig. 8 illustrates that the distinction of enantiomers alone would not be sufficient to explain the change in amplitude of emissions and underlines the putative relevance of introducing storage pools into the models. Underlining the value of a whole ecosystem, holistic approach in the response to the reviewers, the potential implication for a high data requirement for model parameterization remains out of sight. If storage pool dynamics and between species differences in production and storage play an important role, then the holistic approach will tend to give ecosystem/site-specific results. Thus the way to improvement of models, that should be generic, may be more tortuous than imaginable based on merely adding enantiomers.

I excuse if I caused confusion by using the wording „prior to“. My intention was not to suggest waiting with making progress in introducing enantiomers to the models, whenever possible. I only

wanted to point to the need of additional input data for the further development of models.

in the third paragraph of the introduction; The subsequent drought stage → The drought stage
first paragraph after figure 3: it should be not be → it should not be

I suggest to change the x-axis in Ext data Fig. 8a to DOY in order to render comparison to other
figures more easy.

I recommend to check units and underlying biomass in Extended data table 3

Author Rebuttals to First Revision:

Referees' comments:

Referee #1 (Remarks to the Author):

The manuscript has been revised and important with this, previously missing or inadequate information was provided. Further, as the accompanying paper (Werner et al) was now published, it provides a good reference for several background data which otherwise would have been somewhat speculative. Most importantly, the authors now justify the importance of their findings e.g., with the comparison of data with the widely used MEGAN model and show that during the growth the model predictions are poorly matching the data, likely due to complexity in biosynthesis and storage processes related to the enantiomeric distribution of monoterpenes, that changes during drought period.

Thus, I conclude that the ms is now from my perspective recommended for being accepted in Nature.

We thank the reviewer for their comments and their positive final assessment of this work.

Referee #2 (Remarks to the Author):

The authors have addressed many concerns and improved the manuscript quite a bit. Overall the study presents an interesting aspect on the specific nature of certain BVOC emitted from tropical environments. The quantitative nature of the results will likely spur some debate in the analytical community, considering that isoprene concentrations (a compound that should be easy to measure quantitatively in such a setting) differed by a factor of up to ~4 between two complementary analytical techniques during the same campaign.

We thank the reviewer for noting the manuscript has improved based on their comments. We note again that the difference seen in isoprene values between the two isoprene measuring instruments in one time period of the experiment does not affect the conclusions of this paper which concern the enantiomeric monoterpenes, the isoprene signal being used only as a general indicator of de-novo emissions (the same conclusion can be drawn from both datasets). For consistency with the enantiomeric terpene dataset we use the GC-MS data as a comparison. However, we also share the reviewer's frustration that agreement was not perfect throughout. For transparency on this point we now add to the method section the following explanatory text.

“Werner et al. 2021 described the overall ecosystem response to drought and presented data from a PTR-ToF-MS to represent isoprene for consistency with the accompanying soil and cuvette flux data. Here the focus is on enantiomeric monoterpenes which can only be measured by GC-MS techniques as pre-separation is required. Therefore, again for consistency we use the isoprene measured by the same GC-MS instrument. Although broadly similar in the temporal behavior, the

isoprene traces from both systems diverged in concentration during the early drought period (PTR-ToF-MS was lower). Despite rigorous investigation of both systems, no cause for the discrepancy could be found despite the inlets being closely located to each other. Therefore, we concluded that the only remaining plausible cause for the discrepancy is that the sampling lines for the two instruments had differing flow rates, which sampled different locally influenced air. As the temporal behavior of isoprene is used only as an indicator of the general behavior of the de novo emission signal, the short term differences in isoprene concentrations between the instruments are not important in this context, and the same conclusions can be drawn using the other dataset.”

We have also inserted text in the main text which points to the explanation in the methods “(measured by GC-MS see methods for details)”.

Referee #3 (Remarks to the Author):

The revised manuscript essentially addressed my comments and suggestion made during the first round of review and is substantially improved. The merit of the paper is to show the relevance of differences in the dynamics of release of terpene enantiomers from a tropical ecosystem.

I still disagree with the direct line drawn from these findings to model improvement. However, this disagreement is not relevant for the evaluation of the revised paper.

Several aspects remain to be studied in order to gauge the relevance of your findings for modelling. As mentioned in response to reviewer 1 the difference in peaks of light and temperature are less important in the real-world than within the enclosure which probably will reduce the error introduced by a pooled representation of the enantiomers. The comparison of measured concentration to MEGAN flux estimates Fig. 8 illustrates that the distinction of enantiomers alone would not be sufficient to explain the change in amplitude of emissions and underlines the putative relevance of introducing storage pools into the models. Underlining the value of a whole ecosystem, holistic approach in the response to the reviewers, the potential implication for a high data requirement for model parameterization remains out of sight. If storage pool dynamics and between species differences in production and storage play an important role, then the holistic approach will tend to give

ecosystem/site-specific results. Thus the way to improvement of models, that should be generic, may be more tortuouse than imaginable based on merely adding enantiomers.

I excuse if I caused confusion by using the wording „prior to“. My intention was not to suggest waiting with making progress in introducing enantiomers to the models, whenever possible. I only wanted to point to the need of additional input data for the further development of models.

We thank the reviewer for sharing these careful thoughts, in particular regarding the future directions for model studies. The reviewer makes a very good point here, namely that is the real world the error associated with modelling the enantiomers as one compound will be less than for the Biosphere 2. This is because the light and temperature peaks are more separated in the Biosphere 2 than in the Amazon. In actual fact, it is this peculiarity of the Biosphere 2 that made it possible for us to discover the underlying enantiospecific mechanism.

We now use this point to make our discussion of the modelling implications more guarded. We now insert the point as follows “The difference in peaks of light and temperature are less important in the real-world than within the enclosure which will probably reduce the error introduced by a pooled representation of the enantiomers. However, the larger difference between the peaks of light and temperature in the Biosphere 2 enabled for the discovery of the underlying enantiomeric emission differences.”

in the third paragraph of the introduction; The subsequent drought stage → The drought stage
first paragraph after figure 3: it should be not be → it should not be

Done

I suggest to change the x-axis in Ext data Fig. 8a to DOY in order to render comparison to other figures more easy.

Done

I recommend to check units and underlying biomass in Extended data table 3

Done